# The defensome of complex bacterial communities

Angelina Beavogui[1], Auriane Lacroix[1], Nicolas Wiart[2], Julie Poulain[1,3], Tom O. Delmont[1,3], Lucas Paoli [4,5], Patrick Wincker [1,3] & Pedro H. Oliveira [1] ✉

Bacteria have developed various defense mechanisms to avoid infection and killing in response to the fast evolution and turnover of viruses and other genetic parasites. Such pan-immune system (*defensome*) encompasses a growing number of defense lines that include well-studied innate and adaptive systems such as restriction-modification, CRISPR-Cas and abortive infection, but also newly found ones whose mechanisms are still poorly understood. While the abundance and distribution of defense systems is well-known in complete and culturable genomes, there is a void in our understanding of their diversity and richness in complex microbial communities. Here we performed a large-scale in-depth analysis of the defensomes of 7759 high-quality bacterial population genomes reconstructed from soil, marine, and human gut environments. We observed a wide variation in the frequency and nature of the defensome among large phyla, which correlated with lifestyle, genome size, habitat, and geographic background. The defensome's genetic mobility, its clustering in defense islands, and genetic variability was found to be system-specific and shaped by the bacterial environment. Hence, our results provide a detailed picture of the multiple immune barriers present in environmentally distinct bacterial communities and set the stage for subsequent identification of novel and ingenious strategies of diversification among uncultivated microbes.

Bacteria are under constant threat of infection by a variety of genetic parasites such as bacteriophages (henceforth called phages)[1]. As a result of this strong selective pressure, they have evolved multiple sophisticated defense mechanisms capable of regulating the flux of genetic information spread by mobile genetic elements (MGEs) via horizontal gene transfer (HGT)[2–4]. The complete set of bacterial defense systems' repertoire can be designated as their defensome. Several bacterial defense systems have been discovered and extensively discussed in the literature, revealing two major groupings based on their components and

modes of action: innate (non-specific) and adaptive immune systems[5,6]. Typical examples of innate immunity include prevention of phage adsorption[7], restriction-modification (R–M) systems that use methylation to recognize self from non-self-DNA[8], and abortive infection (Abi), in which the infected cell commits suicide before the invading phage can complete its replication cycle[9]. Recent efforts to de-novo identify microbial defense systems resulted in the discovery of several additional innate immune mechanisms with a wide range of genetic architectures[3,4], highlighting the strong selective pressure imposed by genetic

[1]Génomique Métabolique, Genoscope, Institut François Jacob, Commissariat à l'Energie Atomique (CEA), CNRS, Université Evry, Université Paris-Saclay, 2 Rue Gaston Crémieux, 91057 Evry, France. [2]Genoscope, Institut François Jacob, CEA, Université Paris-Saclay, 2 Rue Gaston Crémieux, 91057 Evry, France. [3]Research Federation for the Study of Global Ocean Systems Ecology and Evolution, FR2022 / Tara GOsee, Paris, France. [4]Department of Biology, Institute of Microbiology and Swiss Institute of Bioinformatics, ETH Zürich, Zürich 8093, Switzerland. [5]Institut Pasteur, Université Paris Cité, INSERM U1284, Molecular Diversity of Microbes lab, Paris, France. ✉e-mail: pcoutool@genoscope.cns.fr

parasites on microbial communities. Adaptive immune systems, on the other hand, are so far exclusively represented by clustered, regularly interspaced short palindromic repeats (CRISPR)-Cas, a family of defense systems that provides acquired immunity through the acquisition of short DNA sequences from MGEs that are incorporated into the host genome as spacers[10]. Large-scale efforts for defense system mapping have been recently propelled by the development of bioinformatic tools such as DefenseFinder[11] and PADLOC[12] that rely on a profuse collection of HMM profiles and specific decision rules for each known defense system. Such mapping has been mainly conducted in bacterial species from reference genome databases (e.g., NCBI RefSeq) that are known to overrepresent acute/common human pathogens and organisms that can largely be cultivated in laboratory[11–13]. While extremely insightful, such studies provide a limited snapshot of the bacterial defensome, as they miss the uncharted fraction of environmental microbial diversity that remains uncultured.

The current global Earth microbiome has been estimated at ~$5 \times 10^{30}$ prokaryotic cells[14] scattered throughout a wide range of environments, including deep oceanic and continental subsurfaces, upper oceanic sediment, soil, and oceans as the most densely populated cases. In many environments, 99% of microbes are yet uncultured[15], while cultured representatives belong overwhelmingly to the phyla Actinobacteria, Bacteroidetes, Firmicutes, and Proteobacteria. For nearly 4 billion years, bacteriophages have co-evolved with bacteria, with recent estimates pointing to the presence of ~$10^{31}$ viral particles in the biosphere[16], and up to $10^{23}$ infection events per second taking place just in the global ocean[17].

During the last decade, extensive progress in high-throughput sequencing technologies and computational methods enabled culture-independent genome-resolved metagenomics to recover draft or complete metagenome-assembled genomes (MAGs)[18–20]. The latter have advanced our understanding of the diversity, abundance, and functional potential of microbiota and phageome composition and corresponding ratios across different environments. A healthy adult human gut, for example, is a reservoir for ~$4 \times 10^{13}$ bacterial cells (mostly Firmicutes and Bacteroidetes)[21], and low ($10^{-3}–1$) virus-to-prokaryote ratios (VPRs)[22]. In contrast, marine ecosystems typically show larger VPRs (between $8 \times 10^{-3}–2.15 \times 10^3$, mean of 21.9), followed by soil environments which show the largest ratios (between $2 \times 10^{-3} – 8.2 \times 10^3$, mean of 704) (reviewed in ref. [23]). We hypothesize that the strong VPR dynamics across temporal and spatial scales is likely to profoundly shape the defensome arsenal across biomes.

In this study, we conducted a large-scale in-depth investigation on the abundance, distribution, and diversity of the defensome in complex bacterial communities from three key environments: soil, marine, and the human gut. We tested the association between defensome and different mechanisms of genetic mobility, the former's colocalization in defense islands, and assessed the mutational landscape of high-frequency single nucleotide polymorphisms (SNPs) and insertion-deletions (indels) across defense gene families. These results provide a unique view of the interplay between microbial communities and their phage invaders, and will pave the way to the identification of hitherto unknown defense systems and/or other phage-resistance mechanisms across the enormous diversity of yet-uncultivated microbial populations.

## Results

### Abundance and distribution of defensomes in bacterial MAGs

We performed a defensome mapping across a large dataset of 7759 high-quality (≥90% completeness, ≤5% contamination/redundancy, see Methods) soil, marine, and human gut MAGs[24–26] (Fig. 1a,

Supplementary Data 1–4, and Supplementary Fig. 1). For this purpose, we used a comprehensive collection of hidden Markov model (HMM) protein families and genetic organization rules targeting all major defense system families described in the literature[11] (Fig. 1b).

Throughout this manuscript, we will refer to complete anti-MGE defense systems as those whose currently described genetic organization has been experimentally shown to confer anti-MGE activity. Such a concept of defense system completeness is expected to evolve in the future (particularly for the recently described cryptic large multigenic systems), as more details will emerge regarding their functional intra-operability. In the case of defense genes, they can either belong to complete defense systems, or classify as solitary, i.e., those often shuttled by HGT or arising from genetic erosion of complete defense systems. Of note, the solitary nature of defense genes does not necessarily preclude its functional activity or even implication in anti-MGE defense, as it has been previously shown for solitary bacterial methyltransferases (MTases)[27].

In this study, we found 43,263 defense systems and 764,507 defense genes pertaining to a total of 70 defense families across our full MAG dataset (Supplementary Data 3, 4). The relative distribution of defense systems differed considerably across environments, with R–M, CRISPR-Cas and the SoFIC AMPylase being the most predominant (Fig. 2a). When the distribution of total defense genes was represented instead, we observed multiple solitary genes/incomplete systems (e.g., Gabija, Gao_Qat/Gao_Mza, or Dodola) consistently present across most MAGs (Supplementary Fig. 2). The latter suggests either non-defensive roles or genetic erosion of complete systems similarly to previous observations in complete genomes[13,27]. While defense system distribution across soil and human gut MAGs followed a typical binomial distribution (with most genomes encoding between 3-4 defense systems), that observed in genomes from marine environments was geometric-like, with most MAGs (~65%) showing a limited defensome (Fig. 2b). Such observations are in agreement with recent observations describing a $10^3$ times lower effective rate of HGT in marine bacteria compared with gut bacteria, and with soil bacteria occupying an intermediate position between the former two[28].

Similarly to what has been described for R–M systems[13], we observed positive correlations between the total number of defense systems and MAG size and concomitant negative correlations between the density of defense systems and size (Fig. 2c). Such trends can be explained by the fact that bacteria with larger genomes typically engage in more HGT[2,13], thus requiring a more abundant and diverse defensive arsenal. No qualitative differences were observed when the analyses shown in Fig. 2a–c were performed using MAG assemblies having values of $N_{50} \geq 200$ and 300 kb to control for the effect of contiguity (Supplementary Fig. 3).

The density of defense systems (per MAG and per kb) differed widely among clades, from none (largely in intracellular bacteria and obligatory endosymbionts) to more than $8 \times 10^{-3}$ in *Phascolarctobacterium* (human gut) and ~$1.5 \times 10^{-2}$ in *Elsteraceae* (soil) and UBA9040 (marine) environments (Fig. 2d, Supplementary Fig. 4a, and Supplementary Data 3, 5). No MAG was entirely devoid of defense genes, with maximum densities (per MAG and per kb) ~$8.5 \times 10^{-2}$ across the different biomes (Fig. 2d, Supplementary Fig. 4a, and Supplementary Data 3). When defense systems were split according to its mechanism of action, R–M, Abi, and potential Abi systems were the most prevalent across biomes (Supplementary Fig. 4b and Supplementary Data 6), similarly to recent observations[29].

Apart from MAG size, the abundance of defense genes was expected to depend on phylogenetic depth, as deeper lineages accumulate more events of HGT exchanges, presumably leading to defensome buildup. We ran stepwise linear regression analyses to assess the role of these variables in explaining the variance of the

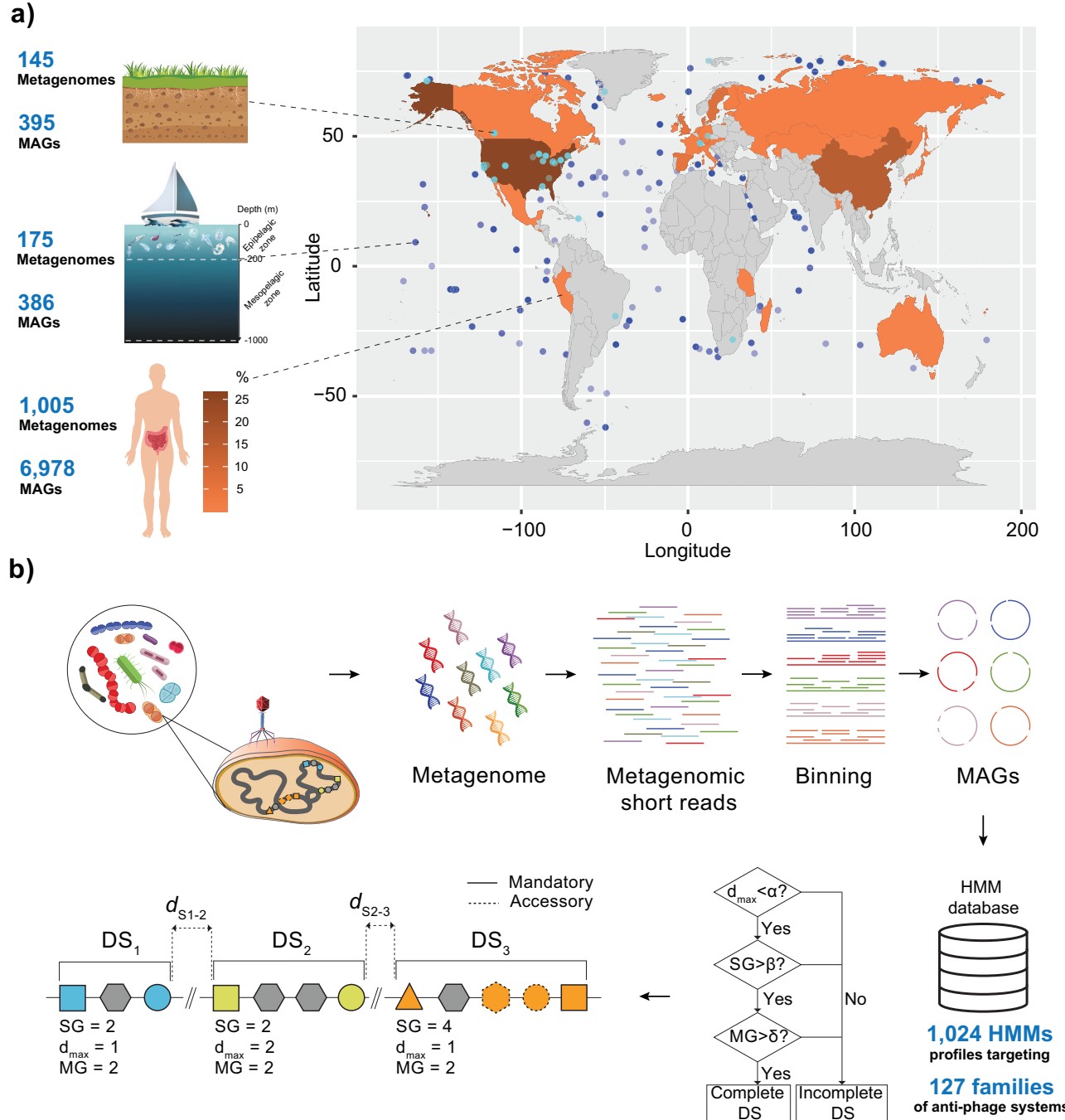

**Fig. 1 | Defensome analysis. a** Our analyses focused on 7759 high-quality near-complete MAGs recovered from three distinct ecosystems: soil, marine, and human gut[24–26]. The geographical distribution of soil and marine sample collection sites is shown in the world map, as well as the percentage of human gut samples recovered from each country (shown as a colored heatmap). Our dataset includes at least 385 Genera (corresponding to a total of 7593 MAGs) and 25 Classes (corresponding to a total of 93 MAGs) not previously covered in a recent study focusing on the defensome of the NCBI RefSeq prokaryotic database[11]. **b** A collection of 1024 HMM profiles targeting 127 families of anti-MGE defense systems from DefenseFinder, was used to query the entire MAG dataset. Briefly, this was performed by means of genetic organization rules allowing for two types of genetic components: "mandatory" and "accessory" (as described previously[11]). Given the wide diversity of genetic organization of anti-MGE systems, rules were written differently for different types of systems. Shown is an example of a genomic region containing three defense systems (DS1-DS3), respectively characterized by a total sum of genes (SG), a maximum distance between defense genes ($d_{max}$), and a given number of mandatory genes (MG), which will allow disentangling between complete or incomplete defense systems based on established thresholds (α, β, δ). Source data are provided as a Source Data file. Image credits (copyright-free) for panels (**a**, **b**): soil (brgfx/Freepik), boat (rawpixel.com/Freepik), plankton (macrovector/Freepik), body/intestines (brgfx/Freepik), bacteria (macrovector_official/Freepik), and phage (Matt Cole/Vecteezy).

defensome (Supplementary Data 7). These showed that MAG size had the strongest direct effect on defensome abundance, and that phylogenetic depth had a significant but less important explanatory role.

We found in our dataset multiple occurrences of ligand binding WYL domains and protein interaction CARD-like domains (Supplementary Fig. 4c–e), with a previously demonstrated regulatory activity toward phage defense systems, namely BREX, CRISPR-Cas, CBASS, and

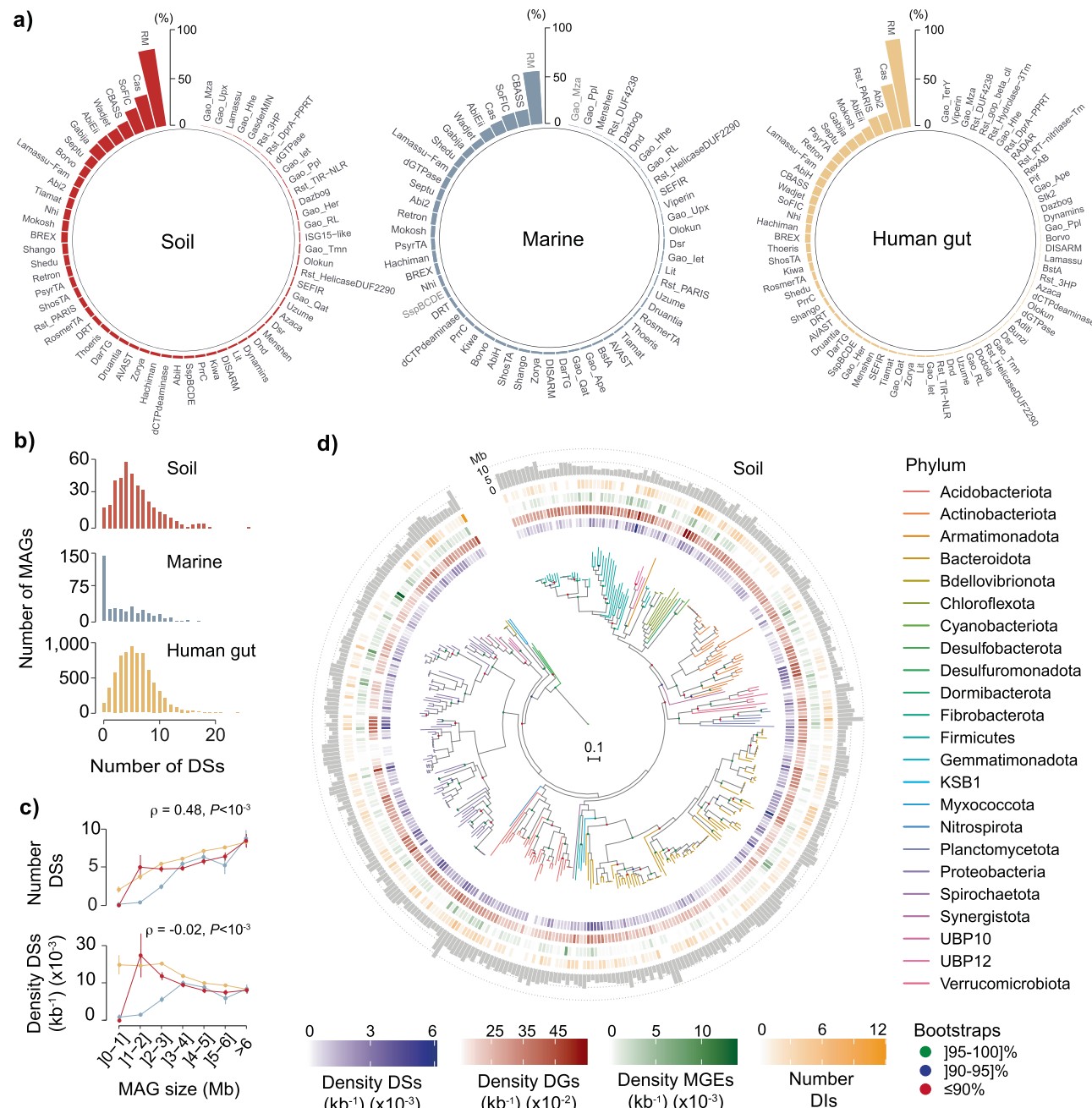

**Fig. 2 | Abundance and distribution of defense systems in MAGs. a** Percentage of soil, marine, and human gut MAGs harboring each family of defense system. **b** Distribution of the number of defense systems (DSs, per MAG) across environments. **c** Variation of number and density (per MAG and per kb) of defense systems (DSs) with MAG size (Mb) for each biome ($n$ = 395, 386, and 6978 MAGs from soil, marine, and human gut environments, respectively). Error bars represent standard deviations of the mean, and correlation was evaluated by a two-sided Spearman's rank test. **d** Phylogenetic representation of 373 soil MAGs, their corresponding phyla, density (per kb) of defense systems (DSs, purple), defense genes (DGs, red), MGEs (green), and number of defense islands (DIs, yellow). The distribution of MAG sizes (Mb) are shown as outer layer barplots. All data corresponds to analyses performed in assemblies with values of $N_{50} \geq 100$ kb. Source data are provided as a Source Data file.

gasdermins[30–33]. Interestingly, we found here a significant colocalization between these domains and multiple defense genes belonging to additional families involved in regulated cell death, such as Lamassu, RosmerTA, and Rst_PARIS. Very few WYL and CARD-like domains were found in genes from marine MAGs (<0.75% of the dataset), in agreement with the latter's more limited defensome. The patterns of colocalization differed across genomes recovered from the soil and human gut environments (Supplementary Fig. 4d–e). For example, WYL preferentially colocalized with CBASS and RosmerTA, respectively, in soil and human gut environments. We also found in the Bacteroidetes

bacterium UBA1952, instances of an operon with some similarity to the recently described *Pedobacter rhizosphaerae* CARD-encoding defense system[33]. In particular, UBA1952 codes for a VapC-like nuclease of the PIN domain superfamily presumably operating as an effector, and the SMC-like RecN with ATPase function (Supplementary Fig. 4c).

Hence, bacterial MAGs possess a diverse repertoire of defense systems (being defense genes essentially ubiquitous), and the patterns of their distribution are very diverse and dependent on genome size and taxonomy. Moreover, defense genes pertaining to systems typically implicated in regulated cell death mechanisms preferentially

colocalize with WYL and CARD-like domains and change according to the environment.

## The interplay between defensome repertoire and bacterial biogeography

Fluctuations in microbial community composition are a function of a large ensemble of diverse biotic and abiotic drivers. Factors such as pH (and other physicochemical parameters), temperature, nutrient availability, or pollution can fundamentally reshape the spatiotemporal dynamics of soil/marine bacterial and viral communities[34–37]. In parallel, multiple variables such as host lifestyle, nutritional needs, genetics, age, medication, urbanization, and the impact of westernization are known to significantly impact the human gut microbiome and virome[38,39]. Concurrent with this dynamic interplay between environmental filtering and phage-bacteria antagonistic and/or mutualistic coevolutionary interactions, one expects concomitant changes in defensome composition. This prompted us to examine how the defensome's abundance and diversity correlated with bacterial biogeography. The top five most represented Classes in our dataset for each environment are Gammaproteobacteria (soil, marine, human gut), Alphaproteobacteria (soil, marine), Dehalococcoidia (marine), Bacteroidia (soil, human gut), and Clostridia (human gut) (Supplementary Data 2). Such different patterns in species richness, and relative phylogenetic diversity across environments, are expected to impact genetic flux and, concomitantly, defensome profiles.

In soil environments, the highest and lowest densities of defense systems were respectively observed in MAGs recovered from serpentine-hosted ecosystems and contaminated or regular soils (Fig. 3a). These observations are consistent with the fact that serpentine environments are among the most challenging niches on Earth, characterized by low cellular abundances, limited microbial diversity, high VPRs[40,41], and consequently, the likely need for additional anti-MGE systems. Conversely, contaminated and regular surface soils impose a type of environmental stress (namely chemical and UV radiation) that is expected to push phage-bacterium interaction from parasitism to mutualism[42–44]. The latter should provide bacterial hosts with diversified competitiveness and environmental adaptability while allowing prophages to avoid direct exposure to the stressor.

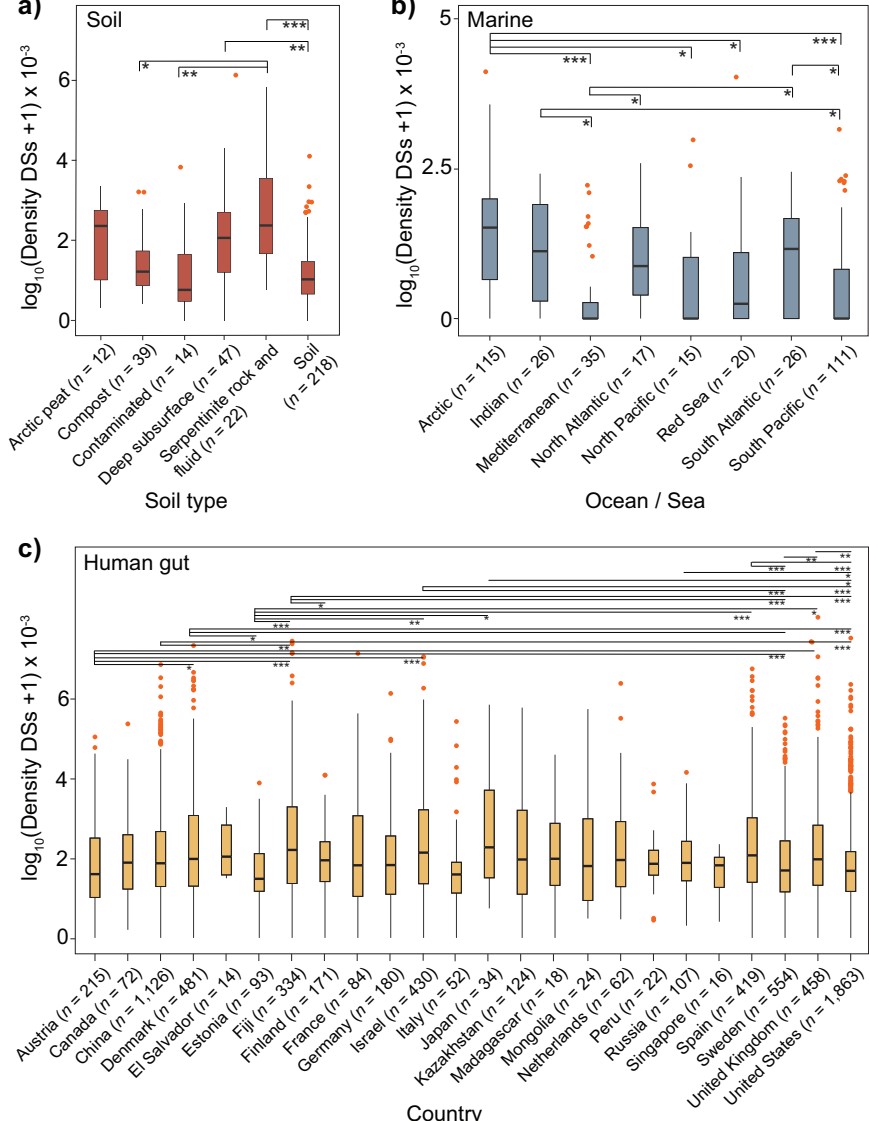

**Fig. 3 | Defensome variation across different ecological and geographical backgrounds.** Defense system (DS) density (per MAG per kb) across distinct ecological (soil, marine) (**a**, **b**) and geographical (human gut) (**c**) contexts. Boxplots represent the 25th to 75th percentiles, the inner black line marks the median, whiskers extend to 1.5x the interquartile range, and data points are set as outliers. Significance was tested by a two-sided Mann–Whitney–Wilcoxon test and P values are indicated as *$P < 0.05$; **$P < 10^{-2}$; ***$P < 10^{-3}$. The number of MAGs analyzed are shown in parentheses. Source data are provided as a Source Data file.

Interestingly, while R–M, CRISPR-Cas, and the less-known standalone SoFIC were prevalent in MAGs recovered from almost all types of soil, arctic peat (richer in Bacteroidales) stands out as an outlier with a high abundance (≥50%) of AbiEii and BREX (Supplementary Fig. 5a, Supplementary Data 8). While it remains unclear which processes drive the overrepresentation of these particular defense families in MAGs recovered from arctic peat, the latter could be explained by the cell's need for a second layer of resistance under conditions of high VPRs (see below), or eventually to enforce cooperation between individuals, or even with MGEs[45,46].

In marine MAGs, we observed the prevalence of R–Ms, but also of the abortive infection system CBASS and SoFIC. The highest defense system densities were found in MAGs originating from the Arctic Ocean (Fig. 3b). Such increased defensive repertoire fits previous observations describing high VPRs and virus-to-bacteria contact rates in sea ice brine compared to seawater[47,48]. Following our observations for ice peat soil, we also found a particularly high abundance (~28%) of the AbiEii system in arctic ocean MAGs (Supplementary Fig. 5b and Supplementary Data 8). The overall low defensome abundance and diversity in the Mediterranean Sea can be due to the latter's conditions of seasonal oligotrophic conditions, higher temperature (>13 °C), and lower concentrations of inorganic nutrients N and P compared to waters of similar depth in open oceans, leading to lower VPRs[49].

To what concerns human gut MAGs, the difference in amplitude in defense system densities across different countries is more subtle (albeit often significant) and harder-to-interpret compared to other environments. While there is a strong trend in the literature supporting a gradual reduction in microbial diversity (and subsequent disruption of metavirome profiles) concomitant with westernization[50], the latter did not translate into a clear-cut geographical trend in regards to the defensome (Fig. 3c and Supplementary Fig. 5c).

When defense systems were split according to its mechanism of action, their variation in density across distinct ecological and geographical backgrounds was kept qualitatively the same, at least for the most abundant mechanisms (R–M, Abi, and potential Abi systems) (Supplementary Fig. 6).

Hence, not only the microbiome but also its defensome is dramatically shaped by different ecological and geographical constraints. Higher densities of defense systems were found in MAGs recovered from particularly challenging biomes such as serpentine soils or the Arctic itself, in line with the high VPRs described in such environments.

## The genetic mobility of bacterial MAG defensomes

Cellular defense genes typically propagate by HGT, in a process frequently mediated by MGEs. Physical colocalization between defense genes and MGEs allows for an efficient strategy to modulate and/or resolve potential conflicts in the interactions between the host and the MGE itself. In this context, a growing number of MGE-encoded defense systems or defense genes have been described in several bacteria, particularly involving the most well-studied ones (R–Ms, Abi, CRISPR-Cas) and major families of MGEs (phages, plasmids, integrons, ICEs/IMEs)[13,51,52]. Yet, there is a paucity of data on the genetic mobility of the defensome in complex bacterial environmental communities. We consistently observed more defense genes in MGEs than in chromosomes (excluding MGEs), irrespective of the environment (Fig. 4a, Supplementary Fig. 7a, and Supplementary Data 9, 10). This is in line with current evidence that MGE-encoded defense systems protect their host cells as a side-effect of their action to protect the MGE from other MGEs[51]. When MGEs were split according to family (excluding integrons which are rare in the human gut microbiota[53]), there was a slight trend for higher colocalization of defense genes with ICEs/IMEs irrespective of the environment (Fig. 4b, Methods), in agreement with recent observations[52]. When integrons were included for comparison, they showed the highest colocalization densities with defense genes in

the human gut (Supplementary Fig. 7b), a result that should be taken cautiously given its low statistical power.

A further split of defense genes according to their corresponding family, allowed us to evaluate the former's over- or under-representation across MGE classes (Fig. 4c and Supplementary Fig. 7c). The results put into evidence a few curious aspects of defensome mobility. The first is that irrespectively of the environment, plasmids generally carry a higher than expected by random chance number of defense genes across a large breadth of defense families when compared to other MGE classes. This observation aligns with the fact that plasmids typically allow for high genetic plasticity and can sustain large gene exchange networks throughout phylogenetically diverse communities[54].

The second aspect relates to the highly heterogeneous landscape of combinations of defense family/MGE class across multiple environments. This reflects the dynamic interplay between a multitude of parameters, including the density and phylogenetic composition of host cells and MGEs present in the community, habitat structure, and environmental pressures. These results also suggest that certain defense genes/systems favor different classes of MGEs for their shuttling, in a likely dynamic and multilayered interplay with shifting allegiances. Overall, these data shows that a wide range of defense families is carried by MGEs, presumably favoring their selfish spread, and that different associations of defense family/MGE class are favored across distinct biomes.

## Encoded functional potential of defense islands and defensome colocalization

Defense genes are typically carried in MGEs by HGT. The former may allow the MGE to be kept in the host by promoting addiction, but on the reverse side of the coin, may carry beneficial traits capable of positive epistatic interactions with the resident host functions. To conciliate these two scenarios, defense genes tend to cluster in so-called defense islands, i.e., high-turnover sinks of genetic diversity, that may serve as catalysts of novel defensive strategies. Therefore, we queried the abundance of such islands and their content. We found 12,890 defense islands in 6217 MAGs (Supplementary Fig. 8a, Supplementary Data 11a, Methods), with a similar size distribution across environments (median ~17 genes) (Fig. 5a), suggesting that there is an optimal size range for these defense sinks. Defense island density was significantly lower in marine environments, followed by soil and human gut (Fig. 5b). The latter is in line with the above observations on a limited defensome in marine MAGs when compared with other environments. Defense islands' anti-MGE content was very diverse (Fig. 5c), with several defense families being overrepresented compared to regions outside defense islands (e.g., Hachiman, R–M, Thoeris) while others being underrepresented (e.g., PsyrTA, ShosTA, Zorya) (Supplementary Fig. 8b).

This bias for certain defense families to locate in defense islands, suggests either positive epistatic interactions with vicinal genes, or a preferential shuttling by a particular family of MGEs. Despite its diversity, defense families are largely similar across environments and skewed towards incomplete systems, pointing as expected, towards a high gene turnover at defense islands (Fig. 5c). Interestingly, the large majority (~63%) of defense islands' gene content was not predicted to have a defensive role. A COG classification of such "non-defensive" genes revealed a high prevalence of functions linked to replication/recombination/repair and transcription (Supplementary Fig. 8c). The latter can be at least partially explained by the fact that defense genes are often shuttled by MGEs, which rely on such functions for target selectivity, insertion, and excision. The above COG categories and the most abundant defense families (R–M and CRISPR-Cas for soil/human gut; R–M, CBASS, and RosmerTA for marine biomes) remained unchanged even when considering defense systems (instead of genes) as the main counting unit in the

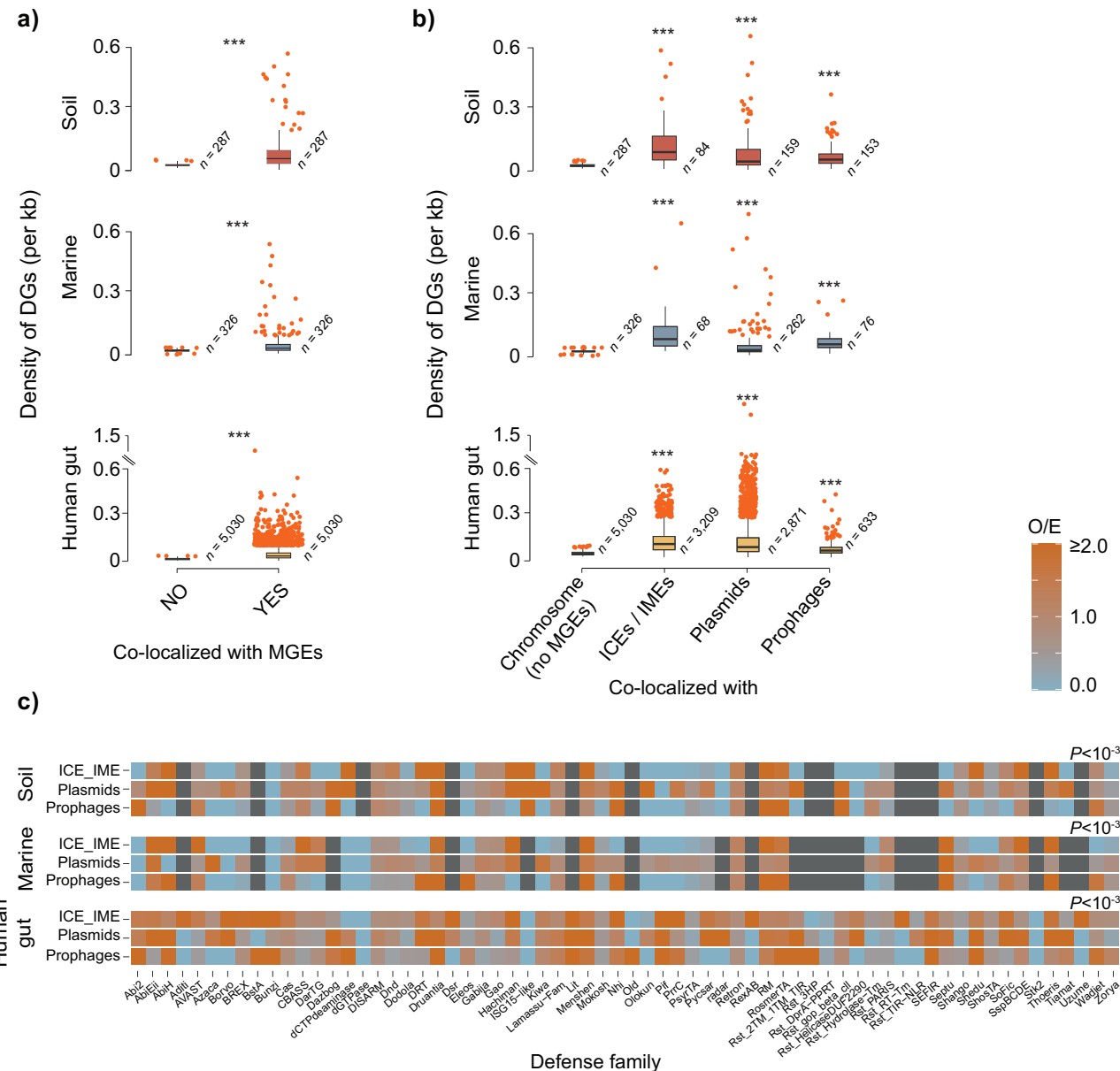

**Fig. 4 | The genetic mobility of the defensome. a** Genomic colocalization of defense genes and MGEs. The number of MAGs analyzed are shown as *n* values. **b** Genomic colocalization of defense genes with plasmids, prophages, and ICEs/IMEs. **c** Heatmap of observed/expected (O/E) ratios of colocalization between genes belonging to distinct defense families and MGEs. Expected values were obtained by multiplying the total number of genes pertaining to a given defense family by the fraction of defense genes of that family assigned to each MGE. Dark gray squares represent the absence of colocalization. Significance was tested by a two-sided Chi-square test. Boxplots represent the 25th to 75th percentiles, the inner black line marks the median, whiskers extend to 1.5x the interquartile range, and data points are set as outliers. Significance was tested by a two-sided Mann–Whitney–Wilcoxon test. *P* values are indicated as ***$P < 10^{-3}$. Source data are provided as a Source Data file.

definition of defense islands (see Methods) (Supplementary Fig. 8d and Supplementary Data 11b).

Since MGEs have different distribution patterns, we quantified the frequency of colocalization of defensome families (≤ 5 genes apart) in defense islands compared to regions outside the latter (Fig. 5d, Supplementary Data 12, Methods). In line with their abundance, frequent shuttling by MGEs and defensive role, R–Ms significantly colocalized with most other defense families in defense islands irrespectively of the environment. Inversely, R–Ms showed a preference to colocalize with genes pertaining to Menshen, Shango, and Dodola families outside defense islands. Interestingly, and despite their general underrepresentation in defense islands (Supplementary Fig. 8b), genes pertaining to families such as PsyrTA and Zorya showed significant colocalization with other defense families inside defense islands. Conversely, certain defense families were significantly overrepresented in defense islands (e.g., Hachiman) (Supplementary Fig. 8b), but rarely colocalized with other families. Upon splitting our dataset according to biogeographical zones, and despite the subsequent decrease in statistical power, the colocalization trends of the most abundant defense families still hold qualitatively (Supplementary Fig. 9 and Supplementary Data 13). These observations point to the possibility of previously unappreciated epistatic interactions between selected families of defense genes/systems in defense islands.

Hence, we found ~11% of the defensome concentrated in defense islands, an environment-dependent highly heterogeneous distribution of defense families in such regions, a large proportion of "non-

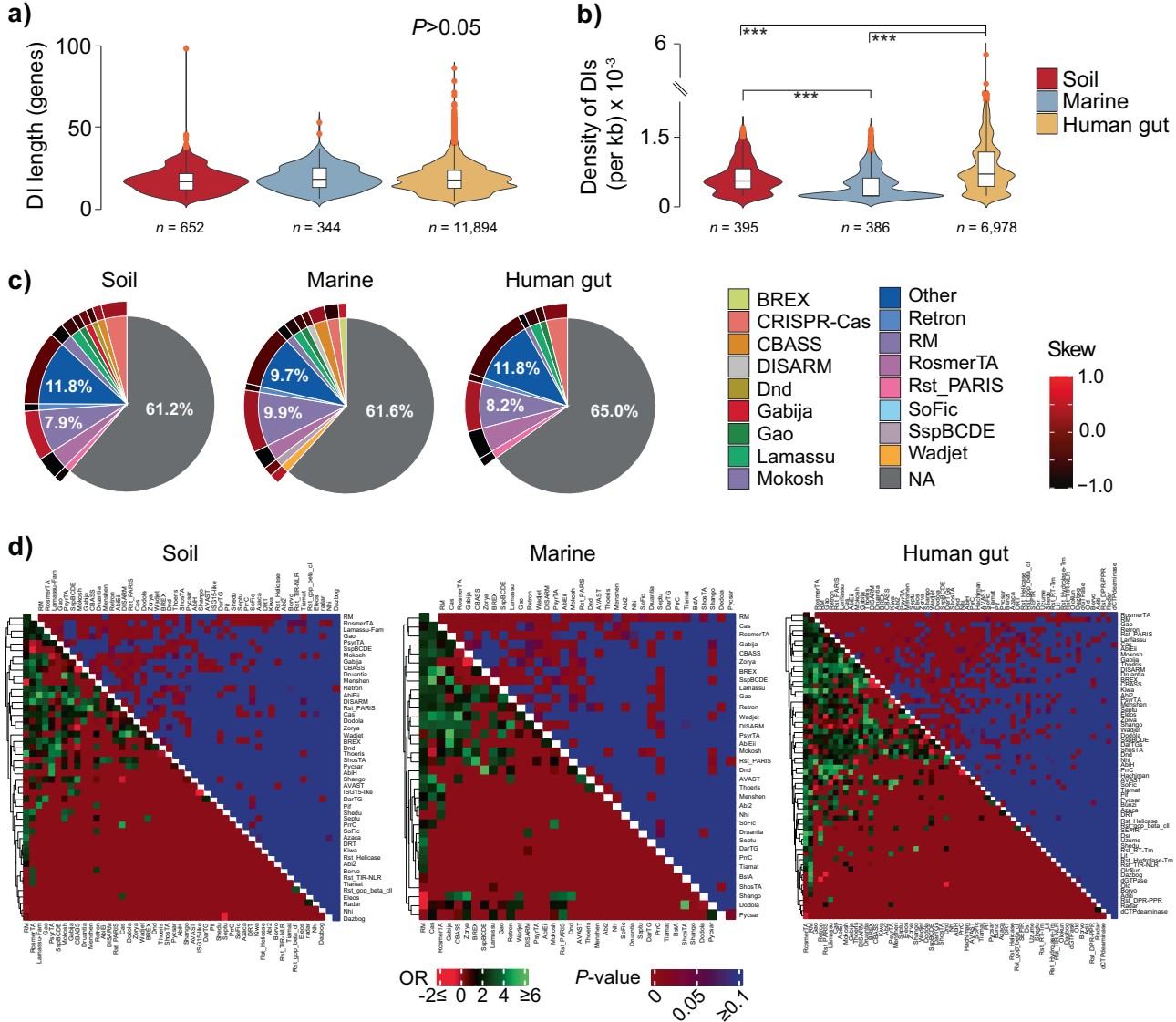

**Fig. 5 | The MAG defense island repertoire. a** Defense island (DI) length distribution (given in genes) in soil, marine, and human gut MAGs. The number of DIs analyzed are shown as *n*. **b** Density distribution of DIs (per MAG per kb) across each environment. The number of MAGs analyzed are shown as *n* values. **c** Pie-plots of the relative abundance (%) of gene content in defense islands. Colored slices correspond to defense genes (those with a relative abundance <1% were merged as 'Other'), and gray slices (NA) correspond to genes not classified as defensive by DefenseFinder. The outer layer corresponds to the skew ratio between genes belonging to complete and incomplete systems given by

$$\frac{\text{\#genes belonging to complete systems} - \text{\#genes belonging to incomplete systems}}{\text{\#genes belonging to complete systems} + \text{\#genes belonging to incomplete systems}}.$$ **d** Defense families'

odds ratio (OR) of colocalization in defense islands (bottom heatmaps) and associated two-sided Fisher's exact test *P* value (upper heatmaps) for the three environments. To eliminate the confounding (inflating) effect of colocalized genes pertaining to the same system, we only considered solitary genes or those pertaining to independent defense systems distanced of 5 genes or less. Boxplots represent the 25th to 75th percentiles, the inner black line marks the median, whiskers extend to 1.5x the interquartile range, and data points are set as outliers. Significance was tested by a two-sided Mann–Whitney–Wilcoxon test. *P* values are indicated as ***$P < 10^{-3}$. Source data are provided as a Source Data file.

defensive" functions, and a significant colocalization of a subset of families of defense.

## The genetic variability of the defensome
The coevolutionary dynamics between defenses and counter-defenses contribute to an endless process of genetic diversification and evolution of sequence specificity, that can take place through point mutations, recombination, gene duplications, replication slippage, or transposition[55]. Such panoply of diversification processes has been particularly studied in well-described innate immune systems like R–Ms, and can take the form of, for example, target recognition domain swapping in Type I *hsdS* subunits, or phase variability of Type III *modH* genes. However, there is a void in our current understanding

of the extent to which differences in selection strength act across distinct defensome gene families. To this end, we performed meta-genome read recruitment over defensome genes, assessed the frequency and type of short variants found, and used this information to pinpoint consistently fast or slow-evolving genes across environments (see Methods for further details).

We observed multiple defense genes with higher-than-expected values of SNP + indel density across multiple biomes (Fig. 6a, only defense families for which at least one defense gene showing an O/E ratio ≥1.5 per environment are shown, Supplementary Data 14). Genes such as *dolB*, *mzaA*, and *sspH* were among this "high-mutation fre-quency" subset irrespective of the environment, while others like *druA*, *zorA*, or *letA* were environment-specific. The results were qualitatively

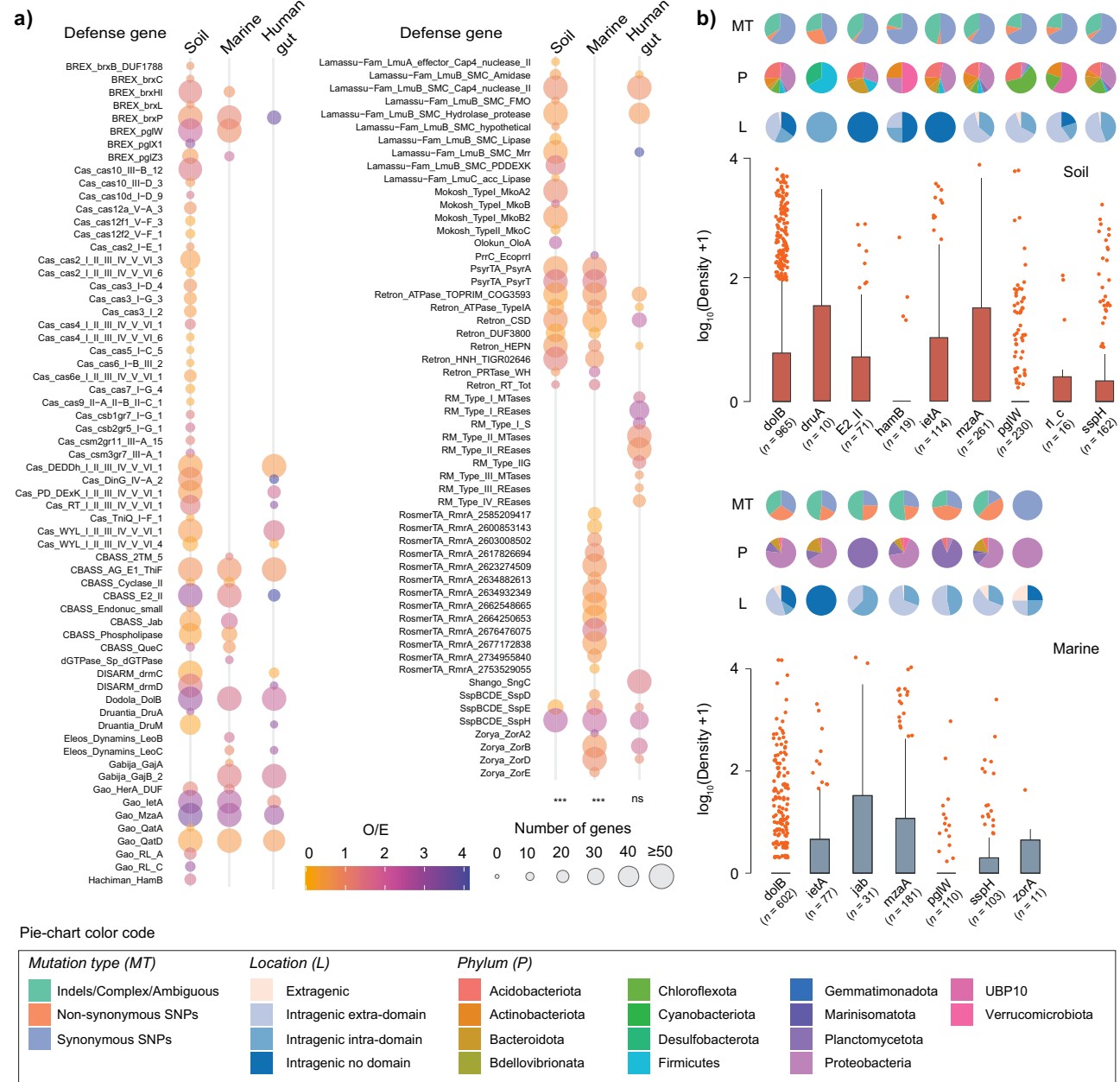

**Fig. 6 | The genetic variability of the defensome. a** Ninety metagenomes (30 for each environment) having a broad representativity in terms of sampling sites (soil and marine) and countries (human gut), as well as in terms of the presence of most defense families previously identified by DefenseFinder were selected. Shown in circles are the observed/expected (O/E) ratios of the number of defense genes harboring high-frequency SNPs + indels (≥ 25% at the variant position) in their gene body (including 200 bp upstream of the start codon). Expected values were obtained by multiplying the total number of genes pertaining to a given defense family by the fraction of defense genes of that family harboring high-frequency alleles. The circle radius corresponds to the total number of defense genes analyzed per family. To ease visualization, we limited the figure to defense families for which at least one defense gene showed an O/E ratio ≥ 1.5 per environment. The complete representation is shown in Supplementary Fig. 10. Significance was tested by a two-sided Chi-square test. **b** Density distribution of SNPs + indels for a selection of defense genes showing the highest O/E values in (**a**). Information on mutation type, location, and phylum are indicated in pie-plots. The number of genes analyzed is shown in parentheses. Boxplots represent the 25th to 75th percentiles, the inner black line marks the median, whiskers extend to 1.5x the interquartile range, and data points are set as outliers. Significance was tested by a two-sided Mann–Whitney–Wilcoxon test. $P$ values are indicated as ***$P < 10^{-3}$; ns not significant. Source data are provided as a Source Data file.

similar when all defense families were included (Supplementary Fig. 10). The range of SNP + indel densities differed considerably across defense gene families (Fig. 6b). Mutation types were also profoundly affected by the environment (and thus by population structure). For example, indels and nonsynonymous SNPs were consistently more abundant in marine than in soil MAGs, even when comparing across same defense genes (e.g., *dolB*, *mzaA*, or *sspH*) (Fig. 6b). While most

variants found was intragenic, *sspH* and particularly *zorA* had as much as 25% of variants located in the first 200 bp upstream the annotated start codons, suggesting potential regulatory effects. The rapid turnover of defense gene repertoires in bacteria, many of which in MGEs, can be followed by selection for the former's conservation or loss in a cell. To investigate the action of natural selection on the defensome gene families showing the highest frequency of variants, we computed

the ratio of nonsynonymous over synonymous substitution rates (dN/dS) for the pools of orthologous defense genes within our MAG dataset. Similar to previous observations for CRISPR-Cas and R–M gene families[13,56], all defense genes analyzed were found to be under strong purifying selection (dN/dS<<1; Supplementary Fig. 11a and Supplementary Data 15). The preferential purge of nonsynonymous mutations by natural selection contributes to maintain the defensive functions of these genes and can be reconciled with a scenario of high turnover, if the selection pressure on the system fluctuates in time, i.e., if these genes alternate periods of strong purifying selection and periods of relaxed selection (e.g., as a result of competition with other defense systems, or during strong selection for HGT). Interestingly, despite their overall negative selection, we observed relatively high levels of divergence and positive selection in certain portions of their sequences (Supplementary Fig. 11b). The latter matched, for example, PFAM domains with predicted AAA+ ATPase activity (PF07724/PF10431 in *dolB*, and to a less extent PF00004/PF17862 in *ietA*), an *ftsH*-like extracellular domain (PF06480 in *ietA*), and a Sigma70-like non-essential domain (PF04546 in *mzaA*).

## Discussion

In this study, we present a large-scale analysis of the abundance and diversity of defensomes of genomes/species from complex microbial communities and three representative biomes: soil, marine, and the human gut. Our results on the quantification of the defensome in marine MAGs lend support to a scenario of a limited defense arsenal in this environment (Fig. 2b). The latter can be accounted by a variety of potential explanations namely: (i) the fact that oligotrophic open oceans typically show an overrepresentation of clades characterized by heavily streamlined genomes[57] (e.g., Dadabacteria, Chloroflexota) (Supplementary Fig 1 and Supplementary Data 2), and thus, more likely to opt for more transient defense systems and little metabolic plasticity to better cope with the limiting environment of the surface ocean; (ii) the dominantly planktonic lifestyle and low cell-density in the marine environment (at least for the free-living fractions accounted for in our MAG dataset) which may in itself, or through a reduced frequency of HGT, contribute to a more limited anti-MGE arsenal; (iii) the fact that the large majority of HMMs currently available to detect defense systems were initially developed on the basis of genetic data that overrepresents not only cultivable bacteria, but also lineages (e.g., *Escherichia*, *Bacillus*, *Pseudomonas*) that are more distantly related to those that make up the global ocean microbiome (Supplementary Data 3). On a broader view, our results qualitatively match those recently obtained for RefSeq genomes in terms of the most abundant systems (R–Ms, CRISPR-Cas) and overall diversity of families identified[11]. The enhanced granularity offered by our cross-environment comparison revealed a few curious differences at the level of preferential 'second line' defense families. One of such differences concerns SoFIC and CBASS which are present in roughly 20% of soil and marine MAGs (mainly in Chitinophagales and Caulobacterales), but considerably less predominant (~8%) in human gut MAGs (mainly in Verrucomicrobiales, Enterobacterales, and Bacteroidales) (Fig. 2a). Inversely, the abortive infection system Rst_PARIS is present in 20% of human gut MAGs (mainly in Bacteroidales) but is virtually absent in soil or marine environments (Fig. 2a). While R–Ms (and to a lesser extent CRISPR-Cas) are largely ubiquitous, our results are supportive of a "second line" of defense systems (SoFIC, CBASS, Rst_Paris, etc.) that is also mostly non-species-specific, differentially favored across distinct environments, and privileged by distinctive strategies of genetic mobility (Fig. 4c, see below). As we move down the ladder of defense system abundance, we face an increasing variety of cryptic, presumably highly specialized, and more species / population-specific systems. By further splitting our dataset into sub-environments or by geographic location, we observed significant differences in defense system

abundance (Fig. 3). And while the increased densities observed at serpentine systems and across the Arctic Ocean can be explained by the extreme conditions experienced at such environments and a subsequent phage-bacteria imbalance, the more subtle variations in defense system density in human gut MAGs across multiple countries and the panoply of confounding variables associated, preclude the identification of more explanatory scenarios.

Higher densities of defense genes were consistently observed in (or in the close vicinity) of MGEs compared to those found in the chromosome (excluding MGEs) (Fig. 4a). Such colocalization facilitates the rapid acquisition and/or diversification of the defensome to provide resistance against multiple other MGEs. It was recently suggested that the carrying of certain defense systems by MGEs by a given bacterial host, may not always relate with the latter's need for protection, but instead, in the best interest of the MGE itself in order to overcome or displace antagonistic MGEs[51]. Our observation of a complex and heterogeneous distribution of defense gene families across different classes of MGEs supports such a hypothesis and suggests an exploitation of MGEs by defense genes/systems for purposes other than host defense. It ultimately highlights the need to better understand the molecular, and evolutionary interactions between the threesome host-phage-mobilome.

Genes acquired by HGT, and MGEs in particular, tend to integrate in a small number of chromosome hotspots to decrease the fitness cost of their integration. Successive rounds of integration/excision/partial deletion of MGEs, when accompanied by the co-option of defense genes/systems, may result in the formation of defense islands. While initially thought that the latter were merely "genomic junkyards" in which the defense genes that are frequently acquired via HGT accumulate because insertion in these regions is unlikely to be deleterious, it has now become clear that there is a specific selective advantage in such clustering of genes, such as functional cooperation between different defensive modules and generation of novel functions. When compared across environments, defense islands did not show significant differences in terms of size, relative abundances of major defense families, or at the topmost abundant COG functional categories for genes classified as 'non-defensive' (Fig. 5a, c and Supplementary Fig. 8c). While many of these genes seem to encode factors involved in genetic mobility, others have hitherto unknown functions. In this line, an interesting next step would be to build upon our precise delimitation of defense islands in such a large and phylogenetically diverse MAG dataset and use a previously developed colocalization framework[3] to leverage the identification of novel defense systems. A significant overrepresentation of several defense families (e.g., Hachiman, R–M, Thoeris) was observed in defense islands (compared to non-island regions). Yet, for certain of these families, such overabundance did not translate into a higher likelihood of colocalization with the remainder of the defensome (and vice-versa). These observations point to the possibility of previously unappreciated epistatic interactions or increased probability of functional diversification for a selected subset of families of defense genes/systems in defense islands. In this regard, the extent to which non-canonical HGT mechanisms (e.g., gene transfer agents, nanotubes, membrane vesicles) and MGE-independent mechanisms of diversification (e.g., homologous recombination) respectively shape the movement of defense genes and the evolution of defense islands remains unclear.

Under the Red Queen evolutionary dynamics, the coevolution between opposing hosts and parasites portrays evolution as a never-ending evolutionary arms-race between defense and counter-defense strategies. Such antagonistic coevolution pervades evolutionary change through multiple ingenious strategies, including: (i) point mutations in phage DNA recognition sites to reduce the likelihood of restriction by R–M systems[58]; (ii) phase-variation/inversions/point mutations in MTases, REases or S modules leading to altered R–M

systems' specificity[55,59]; (iii) ON/OFF switch in CRISPR immunity through mutations in *cas* genes[60]; among others. Thus, not only turnover and recombination, but also rapid sequence evolution of certain defense genes/systems through mutation are key factors shaping the host-parasite evolutionary trajectory. Such diversification strategies are a function of the size and the diversity of the defensome gene pool in a bacterial population, and will shape how the latter remains evolutionarily responsive to temporally or spatially variable selection imposed by phages. Different defense genes are expected to evolve at different rates. For example, significant differences in purifying selection have been described across different Types of R–M REases and MTases, highlighting distinct signatures of adaptive evolution[13]. To gain a birds-eye-view of potentially coexisting sub-populations bearing substantial defense gene-level diversity, we built upon a metagenome read recruitment approach. This allowed us to identify a subset of defense genes having a higher-than-expected frequency of SNPs + indels, globally evolving under strong purifying selection, and a heterogeneous landscape of mutation types profoundly affected by the environment (and thus by population structure). Whereas for some of these genes we can point out determinants capable of explaining such observations—namely the presence of domains known for their predisposition to genetic variation (e.g., the motility-associated *motA* domain[61] in *zorA*, or the *ftsk* translocase domain[62] in *sspH*)—the lack of substantial functional and mechanistic insights on the remaining ones (and on their systems) precludes further meaningful ascertainments.

It is important to appreciate that our computational analysis is challenged by a few difficult-to-control confounding variables and limitations that are worth discussing. The first, concerns the imbalance in our dataset between the number of samples recovered from each biome, as well as their geographic distribution. While the number of soil and marine MAGs analyzed was essentially the same, human gut MAGs were roughly one order of magnitude greater. From the geographic standpoint, marine samples have a global representation, but soil and human gut microbiome data are greatly biased towards the US and China. These observations highlight a critical need for thorough geographic sampling, more global representation of participants in microbiome studies, and fairer access to genomics resources, especially in resource-poor countries. A second confounding variable, likely more relevant, concerns the fact that MAG binning methods using short reads tend to miss certain low-abundance or difficult-to-resolve MGE families. The fact that defense genes are often carried or colocalize with MGEs, necessarily indicates that our results (i) may have a bias in the ratio of defense genes inside versus outside the mobilome, and (ii) are most likely a partial underestimated picture of the real defensome abundance. Future inclusion of long-read data will enable reference-quality genome reconstruction from metagenomes, and further improve our observations. Third, our observations are not representative of all bacterial communities and are likely influenced by the characteristics of the sampled environments. Still, the stringent dataset filtering used in our study in terms of MAG completeness and $N_{50}$ (with associated controls shown in Supplementary Fig. 1), together with a previous demonstration on the accuracy of MAG size estimates (that are part of our dataset) compared with reference genomes[26], makes us have good reasons to think that our analyses constitute a reasonable proxy of the defense landscape diversity carried by such populations, and of the complex interplay underlying their interactions at the intra- and inter-environment level. Lastly, while this study provides novel and intriguing insights into the defensome colocalization, it does not address the specific mechanisms and interactions between different systems, nor the interplay with phage counter-defense strategies[63,64].

The efforts recently undertaken to identify novel defense mechanisms in typically easily cultivable bacteria must now be followed by initiatives to expand the search to uncultivated microbes in complex microbial communities, to understand how such mechanisms collaborate or antagonize with one another, how they co-opt or are co-opted by MGEs, and how they are shaped by the surrounding environment. Our work provides a first stepping stone in such a direction.

## Methods

### Data

In this study, we built upon a large dataset of 7759 high-quality soil, marine, and human gut MAGs[24–26] (Supplementary Data 1). These MAGs were filtered on the basis of the minimum information about a metagenome-assembled genome (MIMAG) standard (≥90% completeness, ≤5% contamination/redundancy, ≥18 tRNA genes, and presence of at least one class of 5S, 16S, and 23S rRNA genes). When not clearly stated in the original study, we performed identification of rRNA genes using both Infernal[65] v1.1.4 (options: -Z 1000 --hmmonly --cut_ga --noali --tblout) and RNAmmer[66] v1.2 (options: -S bac -m tsu,ssu,lsu -h -f -gff) (Supplementary Data 3). Since defense systems are often (i) multigenic and (ii) clustered in defense islands, we further selected for highly contiguous MAGs to more accurately reflect the defensome abundance and distribution. In particular, we selected assemblies having values of $N_{50} \geq 100$ kb (corresponding to at least the top 99.5% best assemblies), and repeated the analyses for $N_{50} \geq 200$ and 300 kb (chosen upon visual inspection of the density distribution) (Supplementary Fig. 1) to account for the effect of contiguity in our observations. MAG annotation was performed with PROKKA[67] v1.14.5 (default parameters).

### Identification of anti-MGE defense genes, systems, and islands

MAGs were queried for anti-MGE defense genes/systems using DefenseFinder[11] v1.0.8 (option: --preserve-raw). The current version of this tool allows for the screening of 1,024 genes pertaining to 127 families of anti-MGE defense systems. Defense islands were defined as arrays of defense genes (or defense systems) separated from one another by ten genes or less and with a minimum of five genes pertaining to at least three different defense families. Functional annotation of "non-defensive" genes was performed with eggNOG-mapper[68] v.2.1.9 (default parameters). To test for colocalization of defense families in defense islands, we computed their odds ratio and associated Fisher's exact test $P$ value. For this purpose, we considered all colocalized defense genes distanced by five genes or less both inside and outside defense islands. Genes belonging to the same defense system are necessarily colocalized, so we deliberately eliminated such hits to avoid inflating the same system colocalization frequencies. To determine the presence of putative defense system regulators harboring WYL or CAspase Recruitment Domains (CARD), all MAG proteomes were scanned against the Pfam-HMMs PF13280 (WYL) and PF00619 (CARD) using HMMER3[69] and a cut-off $e$-value of 0.01.

### Identification of mobile genetic elements

Classification of contigs as belonging to chromosomes or plasmids was performed using PlasClass[70] v.0.1.1 and PlasFlow[71] v.1.1 (both with default parameters). Plasmid hits were selected as those with a score greater than or equal to 0.7. Integrons were identified using IntegronFinder[72] v.2.0.1 (option --local_max). Prophages were detected with Virsorter2[73] v.2.2.3 (options --include-groups dsDNAphage,ssDNA --min-length 5000 --min-score 0.5). Despite recent evidence for phage satellites carrying defense systems[74], we deliberately excluded them from our analyses, mainly due to the very few examples of experimentally validated satellites (particularly in non-cultivable bacteria), which precludes the development of robust detection tools and an accurate evaluation of their classification. Integrative Conjugative Elements (ICEs) and Integrative Mobilizable Elements (IMEs) were detected with ICEfinder[75] v.2.6.32-696.10.2.el6.x86_64 (default parameters). All MGE hits matching multiple families were not considered in the analyses (~2.8% of the

total MGE dataset detected). While MGE carriage by other MGEs (e.g., integrons by plasmids) is indeed expected, we deliberately eliminated such hits to avoid the confounding effects of their co-occurrence on the defensome analyses.

## Phylogenetic analyses

For phylogenetic tree construction we took for each MAG a concatenate of 15 ribosomal proteins (L2, L3, L4, L5, L6, L14, L16, L18, L22, L24, S3, S8, S10, S17, and S19), aligned them with MAFFT[76] v7.490 (options: --maxiterate 1000 -globalpair) (soil, marine) or Muscle[77] v.5.1 (option: -super5) (human gut), and trimmed poorly aligned regions with BMGE[78] v2.0 (option: -t AA). To avoid plotting poorly supported branches, MAGs harboring less than 50% of the abovementioned ribosomal list were omitted from the phylogenetic representations (>95% had the expected number of proteins across the three environments). The trees were computed by maximum likelihood with RaxML[79] v8.2.12 (options: raxmlHPC-PTHREADS-AVX -f a -m PROT-GAMMAAUTO -N autoMRE -p 12345 -x 12345) (soil, marine) or iqtree2[80] v2.2.6 (options: -nt 56 -cmax 15 -bb 1000 -alrt 1000 -m TESTNEW -safe) (human gut) (Supplementary Data 5). The phylogenetic depth was defined as the average root-to-tip distance, and was computed as the diagonal mean of the phylogenetic variance–covariance matrix of each tree, using the vcv.phylo function in the R package "ape".

## Variant analysis of the defensome

To evaluate which defense gene families are preferential targets for increased genetic diversity (SNPs + indels), we selected 90 metagenomes (30 for each environment with similar sequencing depth) having a broad representativity in terms of sampling sites (soil and marine) and countries (human gut), as well as in terms of presence of most defense families that were characteristic to each environment. Fragment recruitment was performed by mapping metagenomic reads from each sample against the ensemble of defense genes (including 200 bp upstream of the start codon) pertaining to the previously selected 90 metagenomes using BWA-MEM[81] v.0.7.17 (default parameters). Genetic variants were identified from aligned reads with FreeBayes[82] v1.1.0 (options: freebayes-parallel -p 1 -P 0 -C 1 -F 0.025 --min-repeat-entropy 1.0 -q 13 -m 60 --strict-vcf -f) and a subsequent filtering step was performed to select only genes (including upstream regions) containing variants having a minimum frequency of 25% supported by at least 10 reads. A minimum of ten genes per defense family per environment was considered in the analysis. Alignments were visualized using IGV v.2.14.1. Finally, SNPGenie[83] v1.0 (options: --vcfformat=4 --snpreport --fastafile --gtffile --outdir) was used for variant classification. For each environment, we computed the observed/ expected (O/E) ratio of defense genes harboring high-frequency alleles across all defense families. Expected values were obtained by multiplying the total number of genes pertaining to a given defense family by the fraction of defense genes of that family harboring high-frequency alleles.

## Analysis of substitution rates

All-against-all BLASTP searches were performed on the sets of defense genes scanned in the genomes (default settings, e-value <$10^{-3}$). Clustering was performed using the SILIX package[84] v.1.3.0 using a minimum identity threshold of 80% and default values for the remaining parameters. Singletons were eliminated from our dataset. The remaining protein sequences (putative orthologs) were reverse-translated to the corresponding DNA sequences using PAL2NAL[85] v14. Pairwise rates of nonsynonymous substitutions (dN), synonymous substitutions (dS), and ω (dN/dS) were computed using the KaKs_Calculator[86] v.2.0 implementing the Yang-Nielsen[87] and Nei-Gojobori[88] methods. Estimations yielding dS >1 (corresponding to situations of substitution saturation and representing 0.2% of the total data) were discarded to improve the quality of the estimation of ω.

## Statistical and graphical analyses of data

All statistical and graphical analyses were conducted using R v.4.3.1. Geographical representation of metagenome sampling locations was generated using the *mapdata* package. Visualization of genomic contexts was performed with the package *gggenes*. Colocalization heatmaps were created using the ComplexHeatmap package. Stepwise linear regression analyses were performed by using the *step* function from the *stats* package.

## Reporting summary

Further information on research design is available in the Nature Portfolio Reporting Summary linked to this article.

## Data availability

All data supporting the findings of this study are available within the article and its supplementary files. Source data are provided with this paper.

## Code availability

Wrapper scripts supporting all key analyses of this work are publicly available at https://github.com/oliveira-lab/Defensome.

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

## Acknowledgements

This work was supported by the Genoscope, the Commissariat à l'Énergie Atomique et aux Énergies Alternatives (CEA), France Génomique (ANR-10-INBS-09–08), and by the Interdisciplinary Center MICROBES of the University Paris-Saclay, as part of the France 2030 program ANR-11-IDEX-0003. L.P. is supported by a European Molecular Biology Organization Postdoctoral Fellowship (EMBO ALTF 100-2023). We thank Eduardo P. C. Rocha (Institut Pasteur, Paris) for his critical reading of the manuscript, and Hadrien Guichard (CEA, Genoscope) for initial efforts in defensome analyses.

## Author contributions

P.H.O. supervised the project. A.B. and P.H.O. designed the computational methods. A.B. and A.L. performed most of the computational analyses and developed most of the scripts that support the analyses. A.B., A.L., N.W., J.P., T.O.D., L.P., P.W., and P.H.O. analysed the data. A.B. and P.H.O. wrote the manuscript with additional information inputs from other co-authors.

## Competing interests

The authors declare no competing interests.
