## [Peer Review File · Nature Communications]

The defensome of complex bacterial communitiesREVIEWER COMMENTS

Reviewer #1 (Remarks to the Author):

It is evident that the study of bacterial immune systems has gained significant traction in the field. In this manuscript, the authors conducted an analysis of immune systems present in bacterial genomes sourced from diverse environments, including soil, marine, and the human gut. The original hypothesis posited that the distinctive nature of these communities would influence the composition and distribution of their respective immune systems.

While this study is certainly of interest, it is imperative to acknowledge a notable limitation stemming from the absence of experimental validation for the primary conclusions drawn in this work. The authors employed well-established, characterized immune systems in their analyses, using them to assess their prevalence within different populations. Assuming the initial hypothesis holds true – that different communities exhibit varying distributions and compositions of immune systems – it is conceivable that these communities may possess additional, as-yet-uncharacterized systems. Drawing from my expertise in phage biology, it is well-established that phages from distinct bacterial species encode uncharacterized immune systems. Keeping this perspective in mind, one could question the extent to which the conclusions proposed in this study hold validity. Given the demonstrated tendency for immune systems to cluster together, it would be valuable to investigate whether uncharacterized proteins encoded by genes within these clusters exhibit anti-phage activity. Failure to do so could potentially undermine the conclusions presented in this work. Additional comments:

1. The authors assert that cellular defense genes typically propagate through horizontal gene transfer (HGT), often mediated by mobile genetic elements (MGEs). However, it is worth noting that many immune systems are located within chromosomal defense islands, and the mechanisms governing their movement remain unclear.
2. Recent research has indicated that satellite phages encode multiple immune systems. It is unclear whether these elements were considered in the analyses conducted in this study.
3. The authors mention a "slight trend for higher colocalization of defense genes with ICEs / IMEs in soil and marine environments, whereas the human gut defensome particularly colocalized with integrons." This finding is surprising, as prophages are more abundant than integrons, and a significant proportion of prophages encode multiple immune systems.
4. The authors also state that "our results on the quantification of the defensome in marine environments lend support to a scenario of a limited defense arsenal." However, this assertion does not take into account, as previously mentioned, the presence of uncharacterized immune systems within these species, which may significantly alter this perspective.

Reviewer #2 (Remarks to the Author):

The study by Beavogui et al. provides a comprehensive analysis of the defensomes in complex microbial communities, shedding light on the variation in frequency and nature of these systems among different phyla and environments. The authors explore the relationship between the defensome and various factors, such as lifestyle, genome size, habitat, and geographic background, providing insights into the evolutionary and ecological drivers of defense system diversity. Importantly, this study provides insights into the defense mechanisms of the majority of prokaryotes, which are uncultured and have been largely understudied. The research presented here is original, interesting and timely and the methodology is sound. Still, the work has a few limitations that should be considered/emphasized:

- The analysis is based on a specific set of high-quality bacterial population genomes reconstructed from soil, marine, and human gut environments. The results may not be representative of all bacterial communities and may be influenced by the specific characteristics of the sampled environments. This should be clearly stated in the penultimate paragraph of the discussion section.
- The study primarily focuses on the abundance and distribution of defense systems and does not provide detailed insights into their functional mechanisms. While the authors mention the genetic

variability and clustering of defense systems in defense islands, the specific mechanisms and interactions between different defense systems are not extensively explored. This should also be clearly stated in the penultimate paragraph of the discussion section.

- The provided methods do not include specific details on the scripts used for the analysis. This can be a limitation for researchers who wish to replicate the study's findings or build upon the work. To address this limitation, the authors could consider sharing their scripts and code (at least those used to produce the main figures of the manuscript and/or the main statistical analyses) in a public repository, such as GitHub, to enhance the reproducibility of their work. This would allow other researchers to access and use the same analysis scripts, facilitating the validation and extension of the study's findings.

- On page 6, the authors state that solitary genes / incomplete systems are consistently present in most MAGs. Is this also true for complete genomes? Just wondering if this might be due to higher fragmentation of MAGs.

- In Figure 4a, I would change the x-axis labels to something like presence or absence of MGEs. As it stands, the sub-figure is only understandable after reading the figure legend.

- It is not clear to me why there is a higher colocalization of defense genes with integrons in the human gut, but integrons carry a lower number of defense genes than expected by chance in the human gut. Please clarify and try to better connect the results shown in Figures 4b and 4c.

- The authors often use the term "anti-phage" to refer to the action of defense systems, but this seems too narrow to me. Is it true that Defense Finder only scans for genes that defend against phages? Recent work (e.g. PMID: 34766904 and PMID: 35388218) has shown that defense systems can target different types of MGEs. Thus, in my opinion, "anti-MGE" would make more sense and be more consistent with the hypothesis that defense systems in MGEs protect their host cells as a side effect of their action to protect the MGE from other MGEs (and not just from phages).

- There seems to be a problem with Supl. Figure 4. The numbers listed in the figure legend don't match those in Figure 1. In addition, panel a) shows marine and human gut, while the figure legends mention soil and human gut. Please correct.

- In Supl. Figure 6, I would change "Ocean" to "Marine" to be consistent.

All the best,
João Botelho

Reviewer #3 (Remarks to the Author):

In this study, the authors used DefenseFinder to analyze metagenome-assembled genomes (MAGs) from three niches, defining the "defensome" of these MAGs in terms of the number and diversity of defense systems, the context of these defense systems within mobile elements and defense islands, defense island content, and patterns of genetic variation of defense systems across different metagenomes. In general, the data presented are interesting. While several studies have analyzed defense system content of bacteria, comparing different phyla and to some extent addressing defense system evolutionary dynamics in different niches, to my knowledge this is the first study providing a niche-resolved analysis of defense machinery from metagenomic sequences. However, for publication, greater depth and care of analysis and contextualization of results would significantly strengthen the manuscript. Comments for the authors' consideration follow:

Major comments:

- The impact of this work would be enhanced by expanding upon three key findings:

i) Better resolution of different defense systems and even defense functions (direct defense vs abortive infection) and their patterns of enrichment across the different niches and subniches studied. E.g. add to figure 3 or SI plots like in figure 2 showing defense systems that were identified in each subniche, and expand discussion about those where systems appeared to be low abundance (e.g. Mediterranean).

ii) Patterns of antiphage system co-occurrence within defense islands (Figure 5D) – given previous observations that systems with complementary functions co-occur (e.g. type I and type IV R-M), this analysis has the potential to provide hints to functions of poorly understood systems, or at least functionally link systems with one another. A more detailed analysis and presentation of these data resolved by niche would be a great benefit to the field (e.g. table in SI showing OR and P-values, genome diagrams and accession numbers, etc).

iii) The genetic variation of system components across ecosystems (Figure 6) has potential to provide significant information on evolutionary pressures existing on different systems and how these differ in different niches. Systems under consistent, or differing mutational pressures should be highlighted, with structural prediction analyses of non-synonymous variant proteins or genomic views of the mutational landscape across promoters, gene bodies, etc to provide the reader with a more in-depth understanding of the meaning of the data presented.

- Many defense systems incorporate relatively conserved and ubiquitous domains in their structures: AAA domains, kinases, and the like. In our experience with DefenseFinder, hits from models to individual genes (what are called here Defense Genes) can thus be somewhat unreliable. This is exemplified in SI figure 2 where several “defense genes” are encoded within 100% of genomes, and the authors point out in these situations that such genes are unlikely to be involved in defense (page 6, line 29–30). Given that the authors observed a skew towards incomplete systems, and a large majority of content that was not defense related within defense islands, I feel this method of identifying defense islands is likely to be too permissive, and is injecting noise into their analysis. The analysis should be performed using a definition that relies on whole systems. In general, the authors should reconsider analyses that rely only on identification of “defense genes” without syntenic evidence for defense function.

- The introduction is basic and should provide more context for the work. No discussion is made around other efforts to delineate the defensomes of other microbial species, as has been undertaken in studies like the one by Tesson et al., which produced the tool DefenseFinder which was a major part of how the authors conducted this study. The authors should present a comprehensive discussion of what this and other papers have revealed regarding the defensome makeup of different species, and thus what might be learned about bacterial immunity by performing a similar analysis on the MAG level.

Minor comments:

- Throughout the manuscript, the authors refer to virus-prokaryote ratios (VPR) calculated in other studies, as a proxy for how hostile a particular environment might be for bacterial cells. The authors should discuss better how these numbers are calculated, particulars of interpreting these numbers, how dynamic they are, and other factors that could explain differences in defensomes in different environments.

- In the intro, superscripts for citations and superscripts for exponents are not well differentiated, leading to some confusion, e.g. page 4 line 10.

- The English throughout the manuscript could use editing – for example page 9 line 14 “being such association environment-dependent.” Also suggest avoiding possessives like in page 9 line 25: “...how the defensome’s abundance...”

- Throughout the manuscript, the authors provide relatively simple interpretations for their results and fail to point out alternative interpretations and discuss better the complexity of the systems they are studying. Several examples follow:

o Page 11: Lines 4–8: I don’t understand the idea here – is the expectation that presence of environmental stress disfavors lytic viruses? A citation is needed here, along with an explanation of other contributing factors for why fewer defense systems were detected in these environments.

o Page 11 lines 10–12: Many defense systems besides AbiEii function through abortive infection, such as CBASS, T-A systems, type III CRISPR, and others – if the authors propose that AbiEii systems are overrepresented in arctic peat due to high VPRs, why aren’t other systems that function via abortive infection enriched too? Why is BREX (which does not have an abi mechanism) enriched?

- o The main explanation for increased defense system density in some environments that the authors provide is high virus-to-prokaryote ratios – what other forces might shape the defenses in these environments and those for which they did not identify many defense systems?
- o Page 14, lines 7–10: Instead of offering the “tantalizing hypothesis” that defense systems tend to be transferred by different mechanisms of HGT, why not provide some plausible explanations here or in the discussion for why these trends were observed?
- o Page 21: lines 10-12: Why would defense system makeup of individual cells be random? Presumably the authors don’t mean to imply here that the cell makes some sort of conscious decision, but that the forces of evolution are at work...?
- In the methods, it states that some Argonautes and Toxin-Antitoxin systems were left out of the analysis. Was the DefenseFinder tool employed as is? Or was the output processed in some way to remove results pertaining to some systems? This should be made more clear.
- Throughout the manuscript, CRISPR systems are referred to as “Cas” – suggested amending to “CRISPR-Cas”
- Figure 2: Is it possible to provide information on how many MAGs for each biome are represented in databases like the ncbi NR database, or the set of microbial genomes analyzed in Tesson et al. where DefenseFinder was first published? This would presumably show that a great deal of diversity in each MAG dataset represents phylogenetic space not previously covered in similar analyses and strengthen the value of this approach – but showing this is the case would strengthen the manuscript.
- Figure 2b: it is interesting that marine samples showed a different distribution of systems compared to other environments – does preponderance of certain species or total diversity of marine MAGs explain this trend? While frequency of HGT may contribute, presumably there may be a variety of reasons here?
- Figure 3: Clarify on figure the number MAGs represented by each box plot – these would be very helpful for interpreting the likely biological significance of the data.
- Figure 4a: “NO” and “YES” labels should be clarified.
- Figure 4c: Suggest coming up with a way to create a unified diagram for all niches – as is, it’s hard to interpret.
- Page 6, Line 29: Gao is referred to as a defense system here and in a couple of other places. DefenseFinder incorporates models for identifying 11 unrelated defense systems identified in Gao et al., 2020 which have the prefix Gao_ in their models (https://github.com/mdmparis/defense-finder-models/blob/master/List_system_article.md) – I don’t think Gao itself is the name of a defense system. Correct to the right system?
- Page 8: line 10-12: Sentence starts out delineating discussion of defense system density by “clade” but ends up discussing by niche. Consider revision.
- Page 8 line 23-25: Authors state that CARD domains have been shown to be transcriptional regulators of antiphage defense systems. While bacterial CARDS are involved in phage defense, my understanding is that they are protein-protein interaction modules and not transcription regulators as suggested here (See Wein et al., citation 31). Unless I’ve missed something, this should be addressed.
- Page 9, Lines 17–20: Needs citation.
- Page 10 lines 5-6: Clarify spelling of serpentinite – sometimes referred to as “serpentine”

Answers to Reviewers

Reviewer 1

Comment 1.0

It is evident that the study of bacterial immune systems has gained significant traction in the field. In this manuscript, the authors conducted an analysis of immune systems present in bacterial genomes sourced from diverse environments, including soil, marine, and the human gut. The original hypothesis posited that the distinctive nature of these communities would influence the composition and distribution of their respective immune systems. While this study is certainly of interest, it is imperative to acknowledge a notable limitation stemming from the absence of experimental validation for the primary conclusions drawn in this work. The authors employed well-established, characterized immune systems in their analyses, using them to assess their prevalence within different populations. Assuming the initial hypothesis holds true – that different communities exhibit varying distributions and compositions of immune systems – it is conceivable that these communities may possess additional, as-yet-uncharacterized systems. Drawing from my expertise in phage biology, it is well-established that phages from distinct bacterial species encode uncharacterized immune systems. Keeping this perspective in mind, one could question the extent to which the conclusions proposed in this study hold validity. Given the demonstrated tendency for immune systems to cluster together, it would be valuable to investigate whether uncharacterized proteins encoded by genes within these clusters exhibit anti-phage activity. Failure to do so could potentially undermine the conclusions presented in this work.

Answer 1.0

We thank the Reviewer for their appreciation of our work and thoughtful suggestions. While we acknowledge that complex environmental bacterial communities likely hold key to so-far uncharacterized proteins with anti-phage activity, their identification and functional characterization is beyond the scope of this study. Given the lack of understanding on the abundance, diversity, mobility, and colocalization of anti-phage defense systems in many of the bacterial species present in such environmental communities, it was in our understanding important to perform this type of large-scale analysis at this point. Although we had already emphasized in our Discussion the great interest of building “upon our precise delimitation of defense islands in such a large and phylogenetically diverse MAG dataset” to “leverage the identification of novel defense systems”, we have now made clear that the latter rests a limitation of this work (**page 26, line 13**): “Lastly, while this study provides novel and intriguing insights into the defensible co-localization, it does not address the specific mechanisms and interactions between different systems, nor the interplay with phage counter-defense strategies.”

In addition, please note that our dataset represents phylogenetic space not previously covered in similar analyses. For example, our study includes at least 385 unique Genera (corresponding to a total of 7,593 MAGs) and 25 Classes (corresponding to a total of 93 MAGs) not previously covered in Tesson et al.¹ (which is the closest most recent study of this kind greatly focusing on cultivable organisms). The former include *Aminicenantia*, *Dormibacteria*, *Kapabacteria*, *Syntrophia*, *Thermoanaerobaculia*, and multiple uncultivated UBAs, which are known to be the first representatives of several major bacterial lineages, substantially expanding genomic representation across the tree of life². We have now added this information to **Fig. 1** caption.

Question 1.1

The authors assert that cellular defense genes typically propagate through horizontal gene transfer (HGT), often mediated by mobile genetic elements (MGEs). However, it is worth noting that many immune systems are located within chromosomal defense islands, and the mechanisms governing their movement remain unclear.

Answer 1.1

This is a good point. While it has indeed been shown that defense genes often propagate by HGT mediated by a broad range of MGEs, and that the latter often cluster in defense islands³, we cannot rule out that additional mechanisms governing their movement are not in action. In addition to the canonical modes of HGT, other mechanisms may operate, for example, gene transfer agents, nanotubes, and membrane vesicles (also termed extracellular vesicles or exosomes). Moreover, we have shown before, that an MGE-independent mechanism - double homologous recombination at flanking core genes -, contributes to diversification of defense islands and beyond⁴. Therefore, we have now added the following sentence to take this point into consideration (**page 24, line 17**): *“In this regard, the extent to which non-canonical HGT mechanisms (e.g., gene transfer agents, nanotubes, membrane vesicles) and MGE-independent mechanisms of diversification (e.g., homologous recombination) respectively shape the movement of defense genes and the evolution of defense islands remains unclear.”*

Question 1.2

Recent research has indicated that satellite phages encode multiple immune systems. It is unclear whether these elements were considered in the analyses conducted in this study.

Answer 1.2

We believe the Reviewer is referring to the study by Rousset and co-workers⁵. While the findings of such study are very interesting and of great scientific relevance, we decided not to include phage satellites in our analyses. The reasons supporting our decision have been recently highlighted⁶, and relate to *i*) the existence of few examples of experimentally validated phage satellites (particularly in non-cultivable bacteria), precluding the development of robust detection tools and an accurate evaluation of their classification, and *ii*) to the fact that satellites can integrate inside other MGEs, notably prophages⁷, adding an additional layer of complexity in their distinction. In any case, we have now clarified in the **Methods** section our option of not including phage satellites in the analysis, and cited appropriate references (**page 28, line 14**).

Question 1.3

The authors mention a "slight trend for higher colocalization of defense genes with ICEs / IMEs in soil and marine environments, whereas the human gut defensome particularly colocalized with integrons." This finding is surprising, as prophages are more abundant than integrons, and a significant proportion of prophages encode multiple immune systems.

Answer 1.3

We thank the Reviewer for pointing this out. We have now deemphasized the integron co-localization data for essentially two reasons: the first relates to the overall low abundance of integrons in intestinal microbiota⁸. The second reason concerns the overall lower statistical power of the integron co-localization dataset when compared with the other MGEs analyzed. In fact, defense genes co-localizing with integrons were found in only 0.1 % of MAGs ($N = 6$) in the

human gut, 5.9 % ($N = 17$) in soil, and 9.8 % ($N = 32$) in marine environments. The corresponding P values of these colocalizations were non-significant ($P > 0.05$) for soil and human gut environments, and only significant ($P < 10^{-3}$) in marine environments. Therefore, we decided to keep in the main figure of the manuscript only the comparisons between MGEs showing higher statistical power (**Fig. 4**), moved to supplementary material (**Supplementary Fig. 7**) the analyses with integrons included, and cautioned for the low statistical power of these results on **page 14, line 8**.

Finally, we have now added sample size (N) values throughout the manuscript (where appropriate) for statistical clarity.

Question 1.4

The authors also state that "our results on the quantification of the defensome in marine environments lend support to a scenario of a limited defense arsenal." However, this assertion does not take into account, as previously mentioned, the presence of uncharacterized immune systems within these species, which may significantly alter this perspective.

Answer 1.4

As mentioned in our **Answer 1.0**, we fully acknowledge that complex environmental bacterial communities most likely hold key to so-far uncharacterized anti-MGE systems. And that several of the species thriving in such communities, will certainly be intensely scrutinized in the next few years with the goal of pinpointing such systems. While it cannot be discarded that previously unappreciated defense systems will indeed end up being identified in such communities, it is less likely that the former will be pervasive across multiple taxa, and even less likely that any of such novel defense families identified, will appear at a frequency rivaling those of R-Ms, CRISPR-Cas, T-As, etc. As it has been shown recently in cultivable bacteria¹, and now in this manuscript, most defense system families remain rare. The latter is likely to also hold true for novel anti-MGE systems and is not expected to abruptly shift the scenario of a limited defensome in marine biomes (or at least its low relative abundance compared with soil and human gut biomes).

Our marine defensome observations are framed based on a large dataset (386 high-quality MAGs) of phylogenetically diverse (pertaining to 183 and 123 unique Genera and Families respectively) communities sampled across 175 locations across the global ocean. While we are confident that our analyses are robust, we have some potential explanations for the limited defense arsenal observed in marine environments:

i) The oligotrophic open oceans are largely dominated by species with heavily streamlined genomes, and thus, less likely to encode defense systems. For example, when compared to soil and human gut MAGs, our marine MAG dataset is: richer in species with genome sizes < 1.5 Mb (12.7% versus 0.25 and 4.8% respectively) (see global distribution in **Supplementary Fig. 1d**); characterized by a larger percentage of genomes with less than 1,500 CDSs (14.7% versus 0.25 and 6.9% respectively) (see global distribution in **Supplementary Fig. 1b**); and enriched in MAGs pertaining to phyla such as Dadabacteria, and Chloroflexota (**Supplementary Table 2**), already shown to exhibit evidence of genome streamlining^{9,10}.

ii) The dominantly planktonic lifestyle and low cell-density in the marine environment (at least for the free-leaving fractions accounted for in our MAG dataset) may in itself, or through a reduced frequency of HGT, contribute to a more limited antiviral arsenal.

iii) One could also argue that the large majority of HMMs currently available to detect defense genes / systems were initially developed on the basis of genetic data that overrepresents not only cultivable bacteria, but also lineages (e.g., *Escherichia*, *Bacillus*, *Pseudomonas*) that are more distantly related to those that make up the global ocean microbiome.

We have now wrapped up all the above in a short paragraph, that we added to the Discussion section (**page 22, line 6**): *“The latter can be accounted by a variety of potential explanations namely: i) the fact that oligotrophic open oceans typically show an overrepresentation of clades characterized by heavily streamlined genomes (e.g., *Dadabacteria*, *Chloroflexota*) (**Supplementary Fig 1, Supplementary Table 2**), and thus, more likely to opt for more transient defense systems and little metabolic plasticity to better cope with the limiting environment of the surface ocean; ii) the dominantly planktonic lifestyle and low cell-density in the marine environment (at least for the free-living fractions accounted for in our MAG dataset) which may in itself, or through a reduced frequency of HGT, contribute to a more limited anti-MGE arsenal; iii) the fact that the large majority of HMMs currently available to detect defense systems were initially developed on the basis of genetic data that overrepresents not only cultivable bacteria, but also lineages (e.g., *Escherichia*, *Bacillus*, *Pseudomonas*) that are more distantly related to those that make up the global ocean microbiome (**Supplementary Table 3**).”*

Additional note to Reviewer 1

In recognition for the careful reading and insightful comments and suggestions, the anonymous Reviewer 1 is now acknowledged in the manuscript.

Reviewer 2

Comment 2.0

The study by Beavogui et al. provides a comprehensive analysis of the defensomes in complex microbial communities, shedding light on the variation in frequency and nature of these systems among different phyla and environments. The authors explore the relationship between the defensome and various factors, such as lifestyle, genome size, habitat, and geographic background, providing insights into the evolutionary and ecological drivers of defense system diversity. Importantly, this study provides insights into the defense mechanisms of the majority of prokaryotes, which are uncultured and have been largely understudied. The research presented here is original, interesting and timely and the methodology is sound.

Answer 2.0

We thank the Reviewer for his appreciation of our work. Please find below detailed answers to the questions/suggestions raised.

Question 2.1

The analysis is based on a specific set of high-quality bacterial population genomes reconstructed from soil, marine, and human gut environments. The results may not be representative of all bacterial communities and may be influenced by the specific characteristics of the sampled environments. This should be clearly stated in the penultimate paragraph of the discussion section.

Answer 2.1

This has now been added to the discussion section (**page 26, line 6**): “Third, our observations are not representative of all bacterial communities and are likely influenced by characteristics of the sampled environments”.

Question 2.2

The study primarily focuses on the abundance and distribution of defense systems and does not provide detailed insights into their functional mechanisms. While the authors mention the genetic variability and clustering of defense systems in defense islands, the specific mechanisms and interactions between different defense systems are not extensively explored. This should also be clearly stated in the penultimate paragraph of the discussion section.

Answer 2.2

We totally agree with the Reviewer. As suggested, we have made this point clear in the discussion section (**page 26, line 13**): “Lastly, while this study provides novel and intriguing insights into the defensome co-localization, it does not address the specific mechanisms and interactions between different systems, nor the interplay with phage counter-defense strategies”.

Question 2.3

The provided methods do not include specific details on the scripts used for the analysis. This can be a limitation for researchers who wish to replicate the study's findings or build upon the work. To address this limitation, the authors could consider sharing their scripts and code (at least those used to produce the main figures of the manuscript and/or the main statistical analyses) in a public repository, such as GitHub, to enhance the reproducibility of their work. This

would allow other researchers to access and use the same analysis scripts, facilitating the validation and extension of the study's findings.

Answer 2.3

We have now included on GitHub (<https://github.com/oliveira-lab/Defensome>) all wrapper scripts enabling to reproduce all key steps of our analyses. When deemed necessary, we also included example input files to guide the users during its deployment. This information has now also been added to the **Methods** section (**page 31, line 5**).

Question 2.4

On page 6, the authors state that solitary genes / incomplete systems are consistently present in most MAGs. Is this also true for complete genomes? Just wondering if this might be due to higher fragmentation of MAGs.

Answer 2.4

This is a good point. The answer is yes, this is also true for complete genomes. We and others have previously observed a large number of solitary / incomplete systems across the bacterial Genbank dataset¹¹. While in some cases such genes have a precise non-defensive functional role (e.g., solitary MTases), a substantial portion of solitary / incomplete systems seems to arise as the result of an ongoing process of genetic erosion of complete systems, at least partially fueled by the elevated genetic turnaround of MGEs carrying them. To make this point clearer, we have now added the following sentence (**page 7, line 31**): *“The latter suggests either non-defensive roles or genetic erosion of complete systems similarly to previous studies in complete genomes”*.

Question 2.5

In Figure 4a, I would change the x-axis labels to something like presence or absence of MGEs. As it stands, the sub-figure is only understandable after reading the figure legend.

Answer 2.5

We have now added the expression “Co-localized with MGEs” on the x-axis of the figure. We also clarified this point in the corresponding figure caption.

Question 2.6

It is not clear to me why there is a higher colocalization of defense genes with integrons in the human gut, but integrons carry a lower number of defense genes than expected by chance in the human gut. Please clarify and try to better connect the results shown in Figures 4b and 4c.

Answer 2.6

We thank the Reviewer for pointing this out. **Figures 4b** and **4c** represent different angles of analysis for this colocalization study. **Figure 4b** compares the density distribution of all defense genes across major classes of MGEs and chromosome (except MGEs). On the other hand, **Figure 4c** gives us a more granular view on the O/E ratio of each defense family in relation to all other defense families (per MGE class). This means that we can have a scenario of multiple underrepresented (low O/E) defense families in a certain MGE class (e.g., integrons), but, if a few defense families are particularly overrepresented, this may lead to a disproportionate distribution in **4b**. Particularly, if N is small. This was exactly the case for defense genes co-localized with integrons in the human gut. The latter were found in only

0.1 % of human MAGs ($N = 6$), compared to 5.9 % ($N = 17$) and 9.8 % ($N = 32$) respectively found in soil and marine environments, and the colocalization was only significant in the latter ($P < 10^{-3}$ in marine environments, but $P > 0.05$ for soil and human gut environments). Therefore, given the low statistical resolution presented by the human gut defense gene / integron dataset, we decided to move the corresponding boxplots and heatmaps to supplementary material (**Supplementary Fig. 7**), and kept in the main figure of the manuscript (**Fig. 4**) only the comparisons with high statistical power. Furthermore, we have cautioned for the low statistical power of these results on **page 14, line 8**. Finally, we have now added sample size (N) values throughout the manuscript (where appropriate) for statistical clarity.

Question 2.7

The authors often use the term "anti-phage" to refer to the action of defense systems, but this seems too narrow to me. Is it true that Defense Finder only scans for genes that defend against phages? Recent work (e.g. PMID: 34766904 and PMID: 35388218) has shown that defense systems can target different types of MGEs. Thus, in my opinion, "anti-MGE" would make more sense and be more consistent with the hypothesis that defense systems in MGEs protect their host cells as a side effect of their action to protect the MGE from other MGEs (and not just from phages).

Answer 2.7

We thank the Reviewer for this observation. Indeed, defense systems target a broad variety of MGEs, and in the recent years we started to see appearing in the literature the term "anti-MGE" more often. We have now changed in the manuscript all occurrences of anti-phage to anti-MGE.

Question 2.8

There seems to be a problem with Supl. Figure 4. The numbers listed in the figure legend don't match those in Figure 1. In addition, panel a) shows marine and human gut, while the figure legends mention soil and human gut. Please correct.

Answer 2.8

Phylogenetic trees were constructed on the basis of a concatenate of 15 ribosomal proteins (L2, L3, L4, L5, L6, L14, L16, L18, L22, L24, S3, S8, S10, S17, S19) for each MAG. To avoid plotting poorly supported branches, MAGs harboring less than 50% of the abovementioned list were omitted from the phylogenetic representations. Please note that such omitted MAGs were still used throughout all our analyses as they fulfilled all the criteria of the MIMAG quality standard. Moreover, > 95% of all our MAG dataset (across the three environments) had the expected number of proteins. We have now reinforced this information in the **Methods** section: "To avoid plotting poorly supported branches, MAGs harboring less than 50% of the abovementioned ribosomal list were omitted from the phylogenetic representations (> 95 % had the expected number of proteins across the three environments)".

Many thanks for catching up the marine / soil typo! It is now corrected.

Question 2.9

In Supl. Figure 6, I would change "Ocean" to "Marine" to be consistent.

Answer 2.9

The modification has been made.

Additional note to Reviewer 2

In recognition for the careful reading and insightful comments and suggestions, the non-anonymous Reviewer 2 Dr. João Botelho is now acknowledged in the manuscript.

Reviewer 3

Comment 3.0

In this study, the authors used DefenseFinder to analyze metagenome-assembled genomes (MAGs) from three niches, defining the “defensome” of these MAGs in terms of the number and diversity of defense systems, the context of these defense systems within mobile elements and defense islands, defense island content, and patterns of genetic variation of defense systems across different metagenomes. In general, the data presented are interesting. While several studies have analyzed defense system content of bacteria, comparing different phyla and to some extent addressing defense system evolutionary dynamics in different niches, to my knowledge this is the first study providing a niche-resolved analysis of defense machinery from metagenomic sequences. However, for publication, greater depth and care of analysis and contextualization of results would significantly strengthen the manuscript. Comments for the authors’ consideration follow:

Answer 3.0

We thank the Reviewer for their summary and appreciation of our work. Please find below detailed answers to the questions/suggestions raised.

Major comments:

Question 3.1

The impact of this work would be enhanced by expanding upon three key findings:

- i) Better resolution of different defense systems and even defense functions (direct defense vs abortive infection) and their patterns of enrichment across the different niches and subniches studied. E.g. add to figure 3 or SI plots like in figure 2 showing defense systems that were identified in each subniche, and expand discussion about those where systems appeared to be low abundance (e.g. Mediterranean).*
- ii) Patterns of antiphage system co-occurrence within defense islands (Figure 5D) – given previous observations that systems with complementary functions co-occur (e.g. type I and type IV R-M), this analysis has the potential to provide hints to functions of poorly understood systems, or at least functionally link systems with one another. A more detailed analysis and presentation of these data resolved by niche would be a great benefit to the field (e.g. table in SI showing OR and P-values, genome diagrams and accession numbers, etc)*
- iii) The genetic variation of system components across ecosystems (Figure 6) has potential to provide significant information on evolutionary pressures existing on different systems and how these differ in different niches. Systems under consistent, or differing mutational pressures should be highlighted, with structural prediction analyses of non-synonymous variant proteins or genomic views of the mutational landscape across promoters, gene bodies, etc to provide the reader with a more in-depth understanding of the meaning of the data presented.*

Answer 3.1

We thank the Reviewer for these extremely thoughtful suggestions. Please find below our answer to the three major requests proposed.

- i) We have now included an additional **Supplementary Fig. 5** and **Supplementary Table 8** depicting the occurrence of each defense system family resolved by biogeographic region across the three biomes. We

have also divided defense families according to their underlying defense mechanism (R–M, Abi, potential Abi, CRISPR-Cas, and other (non-Abi)) and performed the analyses per biome and biogeographic region (**Supplementary Fig. 4b, Supplementary Fig. 6, Supplementary Table 6**). We have made the choice of splitting our dataset according to the classes of mechanisms mentioned above for reasons of uniformity with a recent study¹².

All the above have fueled some considerations that can be found on **page 12, line 18**. Briefly, we stated that while “*it remains unclear which processes drive the overrepresentation of these particular defense families in MAGs recovered from arctic peat, the latter could be explained by the cell’s need for a second layer of resistance under conditions of high VPRs (see below), or eventually to enforce cooperation between individuals, or even with MGEs*”. Also, “*the overall low defensome abundance and diversity in the Mediterranean Sea can be due to the latter’s conditions of seasonal oligotrophic conditions, higher temperature (>13°C), and lower concentrations of inorganic nutrients N and P than waters of similar depth in open oceans, leading to very low VPRs.*” Appropriate references were also added.

- ii) As suggested by the Reviewer, we have now included as additional **Supplementary Fig. 9** and **Supplementary Table 13** the defensome co-localization analyses resolved by biogeographic region across the three biomes. The latter shows that despite the subsequent decrease in statistical power, the colocalization trends of the most abundant defense families still hold qualitatively.
- iii) This is a good point. We now provide as a new **Supplementary Table 14**, detailed information on the location, mutation type, and classification for all variants detected across the ensemble of genes for which we observed a higher than expected by random chance frequency of SNPs + Indels. On top of this, to investigate the action of natural selection on the defensome gene families showing the highest frequency of variants, we computed the ratio of nonsynonymous over synonymous substitution rates (dN/dS) both at whole-gene and per-base level using the Nei-Gojobori, Yang-Nielsen, and Maximum-Likelihood (Akaike Information Criterion) models. Similar to previous observations for R–Ms and Cas gene families, all defense genes analyzed were found to be under strong purifying selection (dN/dS<<1; **Supplementary Fig. 11a**). The preferential purge of nonsynonymous mutations by natural selection contributes to maintain the defensive functions of these genes and can be reconciled with a scenario of high turnover, if the selection pressure on the system fluctuates in time, i.e. if these genes alternate periods of strong purifying selection and periods of relaxed selection (e.g., as a result of competition with other defense systems, or during strong selection for HGT). Interestingly, despite their overall negative selection, we observed relatively high levels of divergence and positive selection in certain portions of their sequences (**Supplementary Fig. 11b,c**). The latter matched, for example, PFAM domains with predicted AAA+ ATPase activity (PF07724 / PF10431 in DoIB, and to a less extent PF00004 / PF17862 in letA), an *ftsH*-like extracellular domain (PF06480 in letA), and a Sigma70-like non-essential domain (PF04546 in MzaA). All this information has now been added to the manuscript (**page 19, line 11**).

Question 3.2

Many defense systems incorporate relatively conserved and ubiquitous domains in their structures: AAA domains, kinases, and the like. In our experience with DefenseFinder, hits from models to individual genes (what are called here Defense Genes) can thus be somewhat unreliable. This is exemplified in SI figure 2 where several “defense genes” are encoded within 100% of genomes, and the authors point out in these situations that such genes are unlikely to be

involved in defense (page 6, line 29–30). Given that the authors observed a skew towards incomplete systems, and a large majority of content that was not defense related within defense islands, I feel this method of identifying defense islands is likely to be too permissive, and is injecting noise into their analysis. The analysis should be performed using a definition that relies on whole systems. In general, the authors should reconsider analyses that rely only on identification of “defense genes” without syntenic evidence for defense function.

Answer 3.2

This is a good point. In our manuscript, defense islands were defined as *arrays of defense genes separated from one another by 10 genes or less and with a minimum of 5 genes pertaining to at least 3 different defense families*. We agree with the Reviewer that such strategy might be somewhat permissive, by allowing the inclusion of genes not involved in defensive roles due to its solitary or incomplete nature. Following the Reviewer’s suggestion, we re-did our analyses, this time redefining defense islands as *arrays of complete defense systems separated from one another by 10 genes or less and with a minimum of 5 genes pertaining to at least 3 different defense families*. As expected, such strategy necessarily led to a dramatic 97.8% reduction in the overall number of defense islands across the three environments. However, despite such reduction, the relative proportion in the number of defense islands across environments was kept tilted to the human gut (~0.5 : 0.3 : 9.2 versus ~1 : 1 : 8 for soil / marine / human gut), the most abundant defense families (R–M and CRISPR–Cas for soil / human gut; R–M, CBASS and RosmerTA for marine biomes) remained unchanged, and the top three functional roles of their non-defensive gene content was also maintained. Hence, redefinition of defense island detection rules, did not change the main conclusions of our analyses. We have now included our additional analyses as **Supplementary Fig. 8d**, and changed the manuscript text to account for these observations (**page 18, line 7**): *“The above COG categories and the most abundant defense families (R–M and CRISPR–Cas for soil / human gut; R–M, CBASS and RosmerTA for marine biomes) remained unchanged even when considering defense systems (instead of genes) as the main counting unit in the definition of defense islands (see **Methods**) (**Supplementary Fig. 8d, Supplementary Table 11b**).”*

Question 3.3

The introduction is basic and should provide more context for the work. No discussion is made around other efforts to delineate the defensomes of other microbial species, as has been undertaken in studies like the one by Tesson et al., which produced the tool DefenseFinder which was a major part of how the authors conducted this study. The authors should present a comprehensive discussion of what this and other papers have revealed regarding the defenseome makeup of different species, and thus what might be learned about bacterial immunity by performing a similar analysis on the MAG level.

Answer 3.3

Due to the rather broad nature of this study, our Introduction section necessarily needs to introduce the reader to a multitude of notions and information that include, among others, the concepts of phage-bacteria arms race, MGE, HGT, adaptive versus innate immune systems, environmental microbial diversity, culture-independent genome-resolved metagenomics, MAGs, etc. Consequently, we had to balance between the diversity of concepts to introduce, depth of detail, and overall length of the section, which in some cases might have resulted in a suboptimal contextualization of the information. While he had already highlighted the importance of the work of Tesson *et al* in the Introduction, we have now added an additional paragraph (**page 3, line 19**), that better describes the current understanding regarding the defenseome makeup in cultivable bacterial species, and the great importance in extending similar analyses on the

MAG level: “Large-scale efforts for defense system mapping have been recently propelled by the development of bioinformatic tools such as DefenseFinder and PADLOC that rely on a profuse collection of HMM profiles and specific decision rules for each known defense system. Such mapping has been mainly conducted in bacterial species from reference genome databases (e.g., NCBI RefSeq) that are known to overrepresent acute / common human pathogens and organisms that can largely be cultivated in laboratory. While extremely insightful, such studies provide a limited snapshot of the bacterial defensesome, as they miss the uncharted fraction of environmental microbial diversity that remains uncultured.”

Minor comments :

Question 3.4

Throughout the manuscript, the authors refer to virus-prokaryote ratios (VPR) calculated in other studies, as a proxy for how hostile a particular environment might be for bacterial cells. The authors should discuss better how these numbers are calculated, particulars of interpreting these numbers, how dynamic they are, and other factors that could explain differences in defensesomes in different environments.

Answer 3.4

This is an important point, and we thank the Reviewer for raising these observations. Per definition, VPR is the ratio of virus-like particles (VLPs) to prokaryotic abundance, and the value that VPR takes will be related to factors controlling both viral and prokaryotic abundance: viral production and decay, prokaryotic production, mortality, and infection rates. While assuming a fixed-rate model for VPR across a given niche remains a suitable first approximation and has been extensively used in the literature, virus abundances are now known to be better described as nonlinear, power-law functions of prokaryote abundances¹³. We have now cautioned for this in the Introduction (**page 4, line 18**): “We hypothesize that the strong VPR dynamics across temporal and spatial scales is likely to profoundly shape the defensesome arsenal across biomes.”

Regarding other factors that might explain differences in defensesomes across different environments, we invite the Reviewer to check our answer to the similar **Question 3.11**.

Question 3.5

In the intro, superscripts for citations and superscripts for exponents are not well differentiated, leading to some confusion, e.g. page 4 line 10.

Answer 3.5

We have now rewritten the sentence to better differentiate both types of superscripts.

Question 3.6

The English throughout the manuscript could use editing – for example page 9 line 14 “being such association environment-dependent.” Also suggest avoiding possessives like in page 9 line 25: “...how the defensesome’s abundance...”

Answer 3.6

We have now rephrased the sentences. Moreover, the manuscript has now been thoroughly proofread to avoid any errors or inconsistencies and allow for style improvement.

Question 3.7

Throughout the manuscript, the authors provide relatively simple interpretations for their results and fail to point out alternative interpretations and discuss better the complexity of the systems they are studying. Several examples follow:

i) Page 11: Lines 4–8: I don't understand the idea here – is the expectation that presence of environmental stress disfavors lytic viruses? A citation is needed here, along with an explanation of other contributing factors for why fewer defense systems were detected in these environments.

ii) Page 11 lines 10-12: Many defense systems besides AbiEii function through abortive infection, such as CBASS, T-A systems, type III CRISPR, and others – if the authors propose that AbiEii systems are overrepresented in arctic peat due to high VPRs, why aren't other systems that function via abortive infection enriched too? Why is BREX (which does not have an abi mechanism) enriched?

iii) The main explanation for increased defense system density in some environments that the authors provide is high virus-to-prokaryote ratios – what other forces might shape the defensome in these environments and those for which they did not identify many defense systems?

iv) Page 14, lines 7–10: Instead of offering the “tantalizing hypothesis” that defense systems tend to be transferred by different mechanisms of HGT, why not provide some plausible explanations here or in the discussion for why these trends were observed?

v) Page 21: lines 10-12: Why would defense system makeup of individual cells be random? Presumably the authors don't mean to imply here that the cell makes some sort of conscious decision, but that the forces of evolution are at work...?

Answer 3.7

We thank the Reviewer for these pertinent observations.

Please find below our detailed answers to the five questions above.

- i) Yes, our claim that certain viral populations and their functions have a profound influence on the adaptation of prokaryotic communities and their ability to withstand adverse geochemical conditions is strongly supported by recent literature. For example, abiotic factors such as resource availability, environmental toxicity, and temperature have all been described as important factors behind this transition from parasitism to mutualism^{14–16}. We have now added supporting references to the manuscript (**page 12, line 13**). In addition, we have provided in the Discussion section, a more in-depth look at additional factors that might be shaping the defensome across niches, particularly regarding our observations on a limited defensome in marine environments (**page 22, line 4**) (see answer to **Question 3.11**).
- ii) This is a good point. It is unclear to us the reasons behind such overrepresentation of AbiE (and to a lesser extent BREX) families in MAGs recovered from arctic peat. Such environments are characterized by anoxia, sub-freezing temperatures, and an abundance of highly promiscuous generalist phages capable of infecting bacterial hosts from different phyla¹⁷. On the other hand, AbiE bicistronic operons function through a Type IV toxin-antitoxin mechanism, which are known to be widespread in bacteria and archaea, and to stabilize MGEs¹⁸. At this point we can only speculate that such extreme conditions favor AbiE either as a second layer of resistance under conditions of high VPRs, or eventually to enforce cooperation between individuals, or even with MGEs. We have now clearly stated that “*it remains unclear which processes drive the overrepresentation of these particular defense families*”, and added the abovementioned references (**page 12, line 18**).

- iii) The amount and diversity of viruses (of which the virus-to-prokaryote ratio is a proxy), the presence of mutualists (prophages and plasmids providing fitness advantages), and the spatial structure of the environment are all predicted to impact prokaryotic immunity (reviewed in ¹⁹). This question is essentially the same as **3.11**, which we now answered in more detail (please see below).
- iv) We have now rephrased the statement as: “*These results also suggest that certain defense genes / systems favor different classes of MGEs for their shuttling, in a likely dynamic and multilayered interplay with shifting allegiances.*”
- v) We were actually referring to a conserved evolutionary relatedness, repeatability and predictability between defense islands, similarly to what has been found in prokaryotic pangenomes²⁰. Yet, we acknowledge that this sentence is still poorly supported by current evidence, and we have now removed it.

Question 3.8

In the methods, it states that some Argonautes and Toxin-Antitoxin systems were left out of the analysis. Was the defensefinder tool employed as is? Or was the output processed in some way to remove results pertaining to some systems? This should be made more clear.

Answer 3.8

We have now removed this sentence. The above-mentioned systems were not included in previous versions of DefenseFinder but are now contemplated in the last version (which was the one used in our final analyses). Thank you for pointing this out.

Question 3.9

Throughout the manuscript, CRISPR systems are referred to as “Cas” – suggested amending to “CRISPR-Cas”

Answer 3.9

We have now changed all occurrences of Cas to CRISPR-Cas.

Question 3.10

Figure 2: Is it possible to provide information on how many MAGs for each biome are represented in databases like the ncbi NR database, or the set of microbial genomes analyzed in Tesson et al. where DefenseFinder was first published? This would presumably show that a great deal of diversity in each MAG dataset represents phylogenetic space not previously covered in similar analyses and strengthen the value of this approach – but showing this is the case would strengthen the manuscript.

Answer 3.10

This is a good point. We would prefer in this case to avoid comparisons with other publicly available MAG datasets essentially for two reasons. First, because to the best of our knowledge, this is the first large-scale study performing this type of defensive analyses on environmental MAGs, and therefore, there simply aren't enough studies out there to perform a reasonable comparison. The second, is that most MAGs published (e.g., those available in ncbi NR), are rather incomplete and/or of low/median quality, and do not fulfill the MIMAG quality criteria that we considered here. Therefore, we find it more logical to compare our study with that of Tesson et al, which is the closest most recent study

of this kind focusing on the defensome of the NCBI RefSeq database (May 2021). In particular we found our study to include at least 385 Genera (corresponding to a total of 7,593 MAGs) and 25 Classes (corresponding to a total of 93 MAGs) not previously covered in Tesson et al. The former include *Aminicenantia*, *Dormibacteria*, *Kapabacteria*, *Syntrophia*, *Thermoanaerobaculia*, and multiple uncultivated UBAs, which are known to be the first representatives of several major bacterial lineages, substantially expanding genomic representation across the tree of life². We have now added this information to **Fig. 1** caption.

Question 3.11

Figure 2b: it is interesting that marine samples showed a different distribution of systems compared to other environments – does preponderance of certain species or total diversity of marine MAGs explain this trend? While frequency of HGT may contribute, presumably there may be a variety of reasons here?

Answer 3.11

There are indeed a variety of potential explanations here:

- i) The oligotrophic open oceans are largely dominated by species with heavily streamlined genomes, and thus, less likely to encode defense systems. For example, when compared to soil and human gut MAGs, our marine MAG dataset is: richer in species with genome sizes < 1.5 Mb (12.7% versus 0.25 and 4.8% respectively) (see global distribution in **Supplementary Fig. 1d**); characterized by a larger percentage of genomes with less than 1,500 CDSs (14.7% versus 0.25 and 6.9% respectively) (see global distribution in **Supplementary Fig. 1b**); enriched in MAGs pertaining to the Dadabacteria, and Chloroflexota phyla (**Supplementary Table 2**), both already shown to exhibit evidence of genome streamlining^{9,10}.
- ii) The dominantly planktonic lifestyle and low cell-density in the marine environment (at least for the free-leaving fractions accounted for in our MAG dataset) may in itself, or through a reduced frequency of HGT, contribute to a more limited antiviral arsenal.
- iii) One could also argue that the large majority of HMMs currently available to detect defense genes / systems were initially developed on the basis of genetic data that overrepresents not only cultivable bacteria, but also lineages (e.g., *Escherichia*, *Bacillus*, *Pseudomonas*) that are more distantly related to those that make up the global ocean microbiome.

We have now wrapped up all the above in a short paragraph, that we added to the Discussion section (**page 22, line 6**): “*The latter can be accounted by a variety of potential explanations namely: i) the fact that oligotrophic open oceans typically show an overrepresentation of clades characterized by heavily streamlined genomes (e.g., Dadabacteria, Chloroflexota) (Supplementary Fig 1, Supplementary Table 2), and thus, more likely to opt for more transient defense systems and little metabolic plasticity to better cope with the limiting environment of the surface ocean; ii) the dominantly planktonic lifestyle and low cell-density in the marine environment (at least for the free-living fractions accounted for in our MAG dataset) which may in itself, or through a reduced frequency of HGT, contribute to a more limited anti-MGE arsenal; iii) the fact that the large majority of HMMs currently available to detect defense systems were initially developed on the basis of genetic data that overrepresents not only cultivable bacteria, but also lineages (e.g., Escherichia, Bacillus, Pseudomonas) that are more distantly related to those that make up the global ocean microbiome (Supplementary Table 3)*”.

Apart from all the above, we added a brief mention in the Discussion, to the fact that our study does not take into account the interplay with phage counter-defense strategies that can disable a variety of bacterial defense mechanisms, and therefore, shape the defensome (**page 26, line 13**): “*Lastly, while this study provides novel and intriguing insights*

into the defensible co-localization, it does not address the specific mechanisms and interactions between different systems, nor the interplay with phage counter-defense strategies.”

Question 3.12

Figure 3: Clarify on figure the number MAGs represented by each box plot – these would be very helpful for interpreting the likely biological significance of the data.

Answer 3.12

We agree with the Reviewer that this is an important point to support the statistical significance of the graphical representations. We have now added the sample number (*N*) of MAGs (or other variables) to each figure throughout the manuscript.

Question 3.13

Figure 4a: “NO” and “YES” labels should be clarified.

Answer 3.13

We have now added the expression “Co-localized with MGEs” to clarify the interpretation of the figure. We also clarified this point in the corresponding figure caption.

Question 3.14

Figure 4c: Suggest coming up with a way to create a unified diagram for all niches – as is, it’s hard to interpret.

Answer 3.14

We agree with the Reviewer. We have now eliminated the redundancy of labels in the xx axis, and fused the three heatmaps in just one, which clearly improves the readability and interpretation of the figure.

Question 3.15

Page 6, Line 29: Gao is referred to as a defense system here and in a couple of other places. DefenseFinder incorporates models for identifying 11 unrelated defense systems identified in Gao et al., 2020 which have the prefix Gao_ in their models (https://github.com/mdmparis/defense-finder-models/blob/master/List_system_article.md) – I don’t think Gao itself is the name of a defense system. Correct to the right system?

Answer 3.15

We thank the Reviewer for pointing this out. We have now precisely split Gao in its 11 defense systems and corrected the sentence accordingly (**page 7, line 28**): “*When the distribution of total defense genes was represented instead, we observed multiple solitary genes / incomplete systems (e.g., Gabija, Gao_Qat / Gao_Mza, or Dodola) consistently present across most MAGs (Supplementary Fig. 2)*”.

Question 3.16

Page 8: line 10-12: Sentence starts out delineating discussion of defense system density by “clade” but ends up discussing by niche. Consider revision.

Answer 3.16

Thanks for pointing this out. We have now changed the sentence to (**page 9, line 14**): “*The density of defense systems (per MAG and per kb) differed widely among clades, from none (largely in intracellular bacteria and obligatory endosymbionts) to more than 8×10^{-3} in Phascolarctobacterium (human gut) and $\sim 1.5 \times 10^{-2}$ in Elsteraceae (soil) and UBA9040 (marine) environments*”.

Question 3.17

Page 8 line 23-25: Authors state that CARD domains have been shown to be transcriptional regulators of antiphage defense systems. While bacterial CARs are involved in phage defense, my understanding is that they are protein-protein interaction modules and not transcription regulators as suggested here (See Wein et al., citation 31). Unless I've missed something, this should be addressed.

Answer 3.17

We thank the Reviewer for spotting this inaccuracy of ours. In fact, WYL domain-containing genes are known to frequently play transcription regulatory functions, but not CARD-containing ones. We have now rephrased the sentence to (**page 10, line 1**): “*We found in our dataset multiple occurrences of ligand binding WYL domains and protein interaction CARD-like domains (**Supplementary Figs. 4c-e**), with previously demonstrated regulatory activity of phage defense systems, namely BREX, CRISPR-Cas, CBASS and gasdermins*”.

Question 3.18

Page 9, Lines 17–20: Needs citation.

Answer 3.18

We have now added appropriate citations.

Question 3.19

Page 10 lines 5-6: Clarify spelling of serpentinite – sometimes referred to as “serpentine”

Answer 3.19

Serpentine soils or *serpentinite soils* are often used interchangeably. We have now adopted the most frequently used form – *serpentine* – throughout the text.

Additional note to Reviewer 3

In recognition for the careful reading and insightful comments and suggestions, the anonymous Reviewer 3 is now acknowledged in the manuscript.

References

1. Tesson, F. *et al.* Systematic and quantitative view of the antiviral arsenal of prokaryotes. *Nat. Commun.* **13**, 2561 (2022).
2. Parks, D. H. *et al.* Recovery of nearly 8,000 metagenome-assembled genomes substantially expands the tree of life. *Nat. Microbiol.* **2**, 1533–1542 (2017).
3. Oliveira, P. H., Touchon, M., Cury, J. & Rocha, E. P. C. The chromosomal organization of horizontal gene transfer in bacteria. *Nat. Commun.* **8**, 841 (2017).
4. Oliveira, P. H., Touchon, M. & Rocha, E. P. C. Regulation of genetic flux between bacteria by restriction–modification systems. *Proc. Natl. Acad. Sci.* **113**, 5658–5663 (2016).
5. Rousset, F. *et al.* Phages and their satellites encode hotspots of antiviral systems. *Cell Host Microbe* **30**, 740–753.e5 (2022).
6. de Sousa, J. A. M., Fillol-Salom, A., Penadés, J. R. & Rocha, E. P. C. Identification and characterization of thousands of bacteriophage satellites across bacteria. *Nucleic Acids Res.* **51**, 2759–2777 (2023).
7. Tommasini, D., Mageeney, C. M. & Williams, K. P. *An integrase clade that repeatedly targets prophage late genes, yielding helper-embedded satellites.* <http://biorxiv.org/lookup/doi/10.1101/2022.07.18.500453> (2022) doi:10.1101/2022.07.18.500453.
8. Buongiorno Pereira, M. *et al.* A comprehensive survey of integron-associated genes present in metagenomes. *BMC Genomics* **21**, 495 (2020).
9. Graham, E. D. & Tully, B. J. Marine Dadabacteria exhibit genome streamlining and phototrophy-driven niche partitioning. *ISME J.* **15**, 1248–1256 (2021).
10. Wiegand, S. *et al.* Taxonomic re-classification and expansion of the phylum Chloroflexota based on over 5000 genomes and metagenome-assembled genomes. *Microorganisms* **11**, 2612 (2023).
11. Oliveira, P. H., Touchon, M. & Rocha, E. P. C. The interplay of restriction-modification systems with mobile genetic elements and their prokaryotic hosts. *Nucleic Acids Res.* **42**, 10618–10631 (2014).
12. Rousset, F. & Sorek, R. The evolutionary success of regulated cell death in bacterial immunity. *Curr. Opin. Microbiol.* **74**, 102312 (2023).
13. Wigginton, C. H. *et al.* Re-examination of the relationship between marine virus and microbial cell abundances. *Nat. Microbiol.* **1**, 15024 (2016).
14. Drew, G. C., Stevens, E. J. & King, K. C. Microbial evolution and transitions along the parasite–mutualist continuum. *Nat. Rev. Microbiol.* **19**, 623–638 (2021).
15. Huang, D. *et al.* Enhanced mutualistic symbiosis between soil phages and bacteria with elevated chromium-induced environmental stress. *Microbiome* **9**, 150 (2021).
16. Tang, X. *et al.* Lysogenic bacteriophages encoding arsenic resistance determinants promote bacterial community adaptation to arsenic toxicity. *ISME J.* **17**, 1104–1115 (2023).
17. Trubl, G. *et al.* Active virus–host interactions at sub-freezing temperatures in Arctic peat soil. *Microbiome* **9**, 208 (2021).
18. Dy, R. L., Przybilski, R., Semeijn, K., Salmond, G. P. C. & Fineran, P. C. A widespread bacteriophage abortive infection system functions through a Type IV toxin–antitoxin mechanism. *Nucleic Acids Res.* **42**, 4590–4605 (2014).
19. Van Houte, S., Buckling, A. & Westra, E. R. Evolutionary ecology of prokaryotic immune mechanisms. *Microbiol. Mol. Biol. Rev.* **80**, 745–763 (2016).
20. Beavan, A., Domingo-Sananes, M. R. & McInerney, J. O. Contingency, repeatability, and predictability in the evolution of a prokaryotic pangenome. *Proc. Natl. Acad. Sci.* **121**, e2304934120 (2024).

REVIEWERS' COMMENTS

Reviewer #1 (Remarks to the Author):

The authors have adequately addressed most of the comments raised in my previous revision. This is a commendable piece of work.

Reviewer #2 (Remarks to the Author):

I would like to thank the authors for promptly incorporating all the suggested revisions. I appreciate their careful attention to feedback, and the changes improve the clarity and quality of the paper. I have no further comments.

Reviewer #3 (Remarks to the Author):

I enjoyed reviewing this manuscript and am pleased my comments were useful in its revision. I now find the revised version suitable for publication without further changes.